# Rad52-Rad51 association is essential to protect Rad51 filaments against Srs2, but facultative for filament formation

**Emilie Ma[1,2], Pauline Dupaigne[2,3,4], Laurent Maloisel[1,2], Raphaël Guerois[2,4,5], Eric Le Cam[2,3,4], Eric Coïc[1,2]\***

[1]DRF, IBFJ, iRCM, CEA, Fontenay-aux-Roses, France; [2]Université Paris-Saclay, Paris, France; [3]Signalisation, Noyaux et Innovation en Cancérologie, Institut Gustave Roussy, CNRS UMR 8126, Villejuif, France; [4]Université Paris-Sud, Orsay, France; [5]DRF, i2BC, LBSR, CEA, Gif-sur-Yvette, France

**Abstract** Homology search and strand exchange mediated by Rad51 nucleoprotein filaments are key steps of the homologous recombination process. In budding yeast, Rad52 is the main mediator of Rad51 filament formation, thereby playing an essential role. The current model assumes that Rad51 filament formation requires the interaction between Rad52 and Rad51. However, we report here that Rad52 mutations that disrupt this interaction do not affect γ-ray- or HO endonuclease-induced gene conversion frequencies. In vivo and in vitro studies confirmed that Rad51 filaments formation is not affected by these mutations. Instead, we found that Rad52-Rad51 association makes Rad51 filaments toxic in Srs2-deficient cells after exposure to DNA damaging agents, independently of Rad52 role in Rad51 filament assembly. Importantly, we also demonstrated that Rad52 is essential for protecting Rad51 filaments against dissociation by the Srs2 DNA translocase. Our findings open new perspectives in the understanding of the role of Rad52 in eukaryotes.

DOI: https://doi.org/10.7554/eLife.32744.001

**\*For correspondence:**
eric.coic@cea.fr

**Competing interests:** The authors declare that no competing interests exist.

## Introduction

Homologous recombination (HR) is a highly conserved mechanism for the repair of DNA double-strand breaks and stalled replication forks. HR defects are associated with many hereditary and sporadic cancers (*Cerbinskaite et al., 2012*), thus underlining the essential nature of this process. HR involves the interaction of the 3'-end of single stranded DNA (ssDNA) with a homologous double-stranded DNA (dsDNA) molecule that is used as a template for DNA synthesis. In eukaryotes, the recombinase RAD51 forms a nucleoprotein filament on ssDNA that performs synapsis and strand invasion of the homologous duplex DNA to form a stable D-loop. The resolution of the joint molecules completes repair (reviewed in *San Filippo et al., 2008*; *Symington et al., 2014*).

RAD51 filament formation on ssDNA coated by the ssDNA-binding protein RPA requires additional factors, such as the mammalian mediator BRCA2 (*Jensen et al., 2010*; *Liu et al., 2010*; *Shahid et al., 2014*; *Thorslund et al., 2010*; *Zhao et al., 2015*) or the yeast protein Rad52 (reviewed in *Symington et al., 2014*). In yeast, Rad52 is the most important mediator of Rad51 filament formation and it was proposed that this function involves its interaction with both RPA and Rad51. A domain essential for interaction with RPA was identified (*Plate et al., 2008*) and two motifs, FVTA at position 316–319 and YEKF at position 376–379, are each essential for interaction with Rad51 (*Krejci et al., 2002*; *Kagawa et al., 2014*) (*Figure 1a*). In mammalian cells, RAD51 filament formation is mainly driven by BRCA2 and the role of RAD52 is still enigmatic (*Liu and Heyer, 2011*). RAD52 is required for the viability of BRCA2-deficient cells (*Feng et al., 2011*), underlining the

importance of its function. Recent studies revealed a role for RAD52 pairing activity in promoting DNA synthesis following replication stress (*Bhowmick et al., 2016*; *Sotiriou et al., 2016*). However, RAD52 interaction with RPA and RAD51 (*Park et al., 1996*; *Shen et al., 1996*) suggests that RAD52 has a role in RAD51 filament regulation also in mammalian cells.

It was shown in yeast that the DNA translocase Srs2 is required to avoid the accumulation of toxic Rad51 filaments. Specifically, HR is accountable for the sensitivity of Srs2-deficient cells to DNA damage. As Srs2 can dismantle Rad51 filaments in vitro, the accumulation of toxic Rad51 filaments in Srs2-deficient cells is probably responsible for this sensitivity (*Burgess et al., 2009*; *Esta et al., 2013*; *Krejci et al., 2003*; *Veaute et al., 2003*). The unproductive association of Rad51 with ssDNA could interfere with the normal progression of DNA replication forks or DNA repair (*Aboussekhra et al., 1992*; *Klein, 2001*; *Vasianovich et al., 2017*). However, this control of toxic filaments suggests that functional Rad51 filaments are protected from Srs2 activity. The Rad55/57 and SHU complex, which contains Rad51 paralog proteins, was shown to provide such protection (*Bernstein et al., 2011*; *Fung et al., 2009*; *Liu et al., 2011*).

In several organisms, RAD51 paralogs also stabilize RAD51 filaments by inhibiting RAD51 dissociation from ssDNA (*Taylor et al., 2016*; *Da Ines et al., 2013*; *Liu et al., 2011*), thus ensuring the success of homology search. Rad52 might also share this role as shown by its association with Rad51 filaments after their formation (*Esta et al., 2013*). In this study, we proposed that this association is responsible for the toxicity of Rad51 filaments in Srs2-deficient cells because Rad52 mutations that do not affect Rad51 filament formation suppress the sensitivity of Srs2-deficient cells to DNA-damaging agents. We also found that *SIZ2* overexpression promotes massive Rad52 sumoylation and leads to the suppression of Rad51 filament toxicity in Srs2-deficient cells. Therefore, we hypothesized that Rad52 sumoylation could trigger Rad52 dissociation from Rad51 filaments (*Esta et al., 2013*).

We sought to better understand the role of Rad52 in the formation of toxic Rad51 filaments. To this aim, we used a random mutagenesis approach to select new *RAD52* gene mutations that suppress the sensitivity of Srs2-deficient cells to DNA damage. We found mutations that restrict the interaction between Rad52 and Rad51, showing that this interaction is responsible for Rad51 filament toxicity in Srs2-deficient cells. Unexpectedly, our genetic and molecular analyses show that the interaction between Rad52 and Rad51 is not absolutely required for Rad51 filament assembly. Additionally, we found that in Srs2-proficient cells, Rad52 is required to protect Rad51 filaments from disruption by Srs2. Altogether, our results bring a different view on the regulation of Rad51 filaments by Rad52 and Srs2. We propose a unified model to explain the crucial roles of Rad52 and Srs2. The newly identified Rad52 role could be extended to other eukaryotes and to other mediator proteins.

## Results

### Identification of *rad52* mutants that suppress Rad51 filament toxicity in Srs2-deficient cells

To better understand the role of Rad52 in Rad51 filament toxicity in Srs2-deficient cells, we searched for *rad52* mutants that upon DNA damage, suppress the lethality induced by Rad51 toxic filaments generated in the absence of Srs2. To this aim, we screened a *RAD52* randomly mutated plasmid library to identify mutants that suppress the sensitivity of cells lacking *RAD52* and *SRS2* (*rad52Δ srs2Δ*) to high dose of methyl methanesulfonate (MMS, 0.015%). We used cells lacking both *SRS2* and *RAD52* to find *RAD52* mutations that complement the essential functions of this gene in Rad51 filament formation. To confirm that the suppression phenotype of the selected mutants was only caused by the mutation carried by the selected plasmids and not by background mutations, we isolated each plasmid from these strains and used them to transform again *rad52Δ srs2Δ* cells. Evaluation of resistance to MMS by spot assay (*Figure 1—figure supplement 1a*) showed that each isolated plasmid significantly restored MMS resistance in *rad52Δ srs2Δ* cells. The list of selected mutations is in *Supplementary file 1*. Particularly, we found mutations in three consecutive residues, $F_{379}$ $A_{380}$ $P_{381}$ (*Figure 1b*), that define the FAP domain. $F_{379}$ belongs to the YEKF motif at position 376–379 (*Figure 1a*), the deletion of which strongly impairs Rad52 binding to Rad51 (*Krejci et al., 2002*). We chose for the subsequent analyses a representative FAP mutant (*rad52-P381S*) because it strongly reduced sensitivity to MMS in Srs2-deficient cells (*Figure 1c*). The strong suppression was

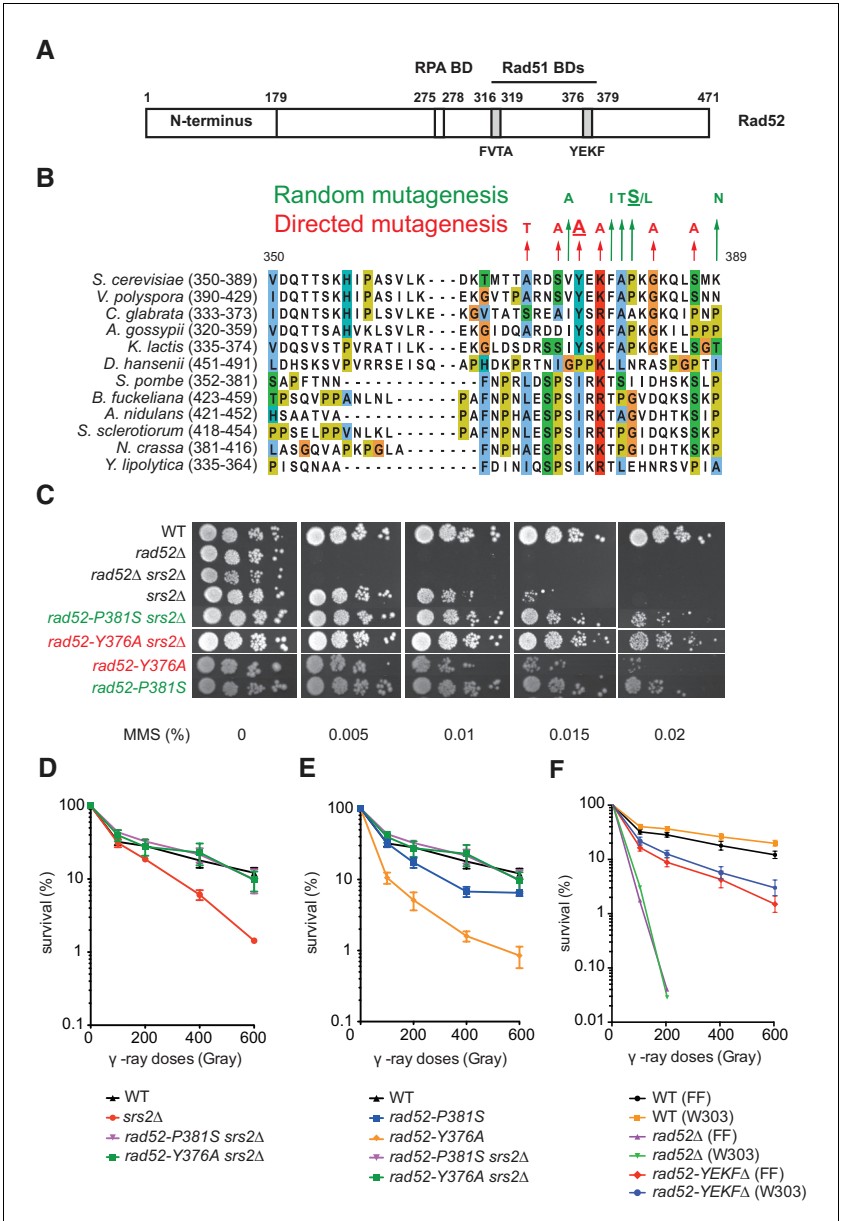

**Figure 1.** Mutations in the Rad51-binding domain of Rad52 suppress MMS and γ-ray sensitivity of Srs2-deficient cells. (A) Primary structure of Rad52. The conserved N-terminus as well as the RPA and the Rad51 binding domains (BD) are shown. (B) The mutations selected from random and directed mutagenesis are indicated on the alignment of the region 350-389aa of *S. cerevisiae* Rad52 with that of Rad52 orthologs in 12 fungal species. The amino acid colors refer to the chemical character of the residues. The *rad52-P381S* and *rad52-Y376A* mutations are underlined. (C) Serial 10-fold dilutions of haploid strains with the indicated genotypes were spotted onto rich media (YPD) containing different MMS concentrations. The data presented are a selection from those in *Figure 1—figure supplement 1*. (D,E,F) Survival curves of haploid cells exposed to γ-ray. *rad52-YEKFΔ* cells were from two different genetic backgrounds: FF18733 (FF) and W303. Data are presented as the mean ± SEM of at least three independent experiments.

DOI: https://doi.org/10.7554/eLife.32744.002

The following source data and figure supplements are available for figure 1:

**Source data 1.** Survival (%) of haploid cells exposed to γ-ray for *Figure 1D,E and F*.
DOI: https://doi.org/10.7554/eLife.32744.005

**Figure supplement 1.** Mutations in the Rad51-binding domain of Rad52 suppress MMS and γ-ray sensitivity of Srs2-deficient cells.

*Figure 1 continued on next page*

*Figure 1 continued*

DOI: https://doi.org/10.7554/eLife.32744.003

**Figure supplement 2.** The *rad52-Y376A*, but not the *rad52-P381S* mutation suppresses the effect of mutations that are synthetically lethal with *srs2Δ*.

DOI: https://doi.org/10.7554/eLife.32744.004

confirmed by the finding that the survival rate of *rad52-P381S srs2Δ* cells exposed to γ-rays was comparable to that of wild type (WT) cells (*Figure 1d*).

## The Rad52 YEKF domain of interaction with Rad51 is involved in Rad51 filament toxicity, but is dispensable for HR

The fact that $F_{379}$ belongs to the YEKF domain involved in Rad52 binding to Rad51 (*Krejci et al., 2002*) suggests that the interaction between the two proteins is effectively involved in Rad51 filaments toxicity. To test further this correlation, we measured the MMS and γ-ray sensitivity of Srs2-defective cells carrying a mutation in one of the six well-conserved amino acids identified by sequence alignment in or around the YEKFAP domain of 12 Rad52 orthologs in fungi (*Figure 1b*). First, we assessed the capacity of each mutation to suppress MMS sensitivity of *rad52Δ srs2Δ* cells by spot assay. We found that *rad52-Y376A* fully suppressed the MMS and also γ-ray sensitivity of Srs2-deficient cells (*Figure 1c,d*), whereas the effect of the *rad52-K378A* and *rad52-G383A* mutations was less strong (*Figure 1—figure supplement 1b,d*). This mutational analysis suggests a link between the Rad51-binding domain in Rad52 and the potential toxicity of Rad51 filaments. Accordingly, crossing *rad52-Y376A srs2Δ* haploid cells with DNA repair or replication haploid mutants, such as *rad50Δ*, *rad54Δ*, *rrm3Δ*, *mrc1Δ*, *ctf18Δ* and *sgs1Δ*, also suppressed the synthetic sickness or lethality observed in the absence of Srs2 (*Figure 1—figure supplement 2*) and attributed to the accumulation of toxic Rad51 filaments (*Esta et al., 2013*). The *rad52-P381S* allele could not suppress lethality, indicating that its effect on Rad51 filaments toxicity is less important than that of *rad52-Y376A*.

To assess the effect of Rad51-binding domain mutations on Rad52 function, we transformed Rad52-deficient cells (but proficient for Srs2) with plasmids harboring *RAD52* mutations. Most of them restored almost completely resistance to genotoxic agents in *rad52Δ* cells (*Figure 1—figure supplement 1c,e*). However, the *rad52-Y376A* mutant did not fully rescue the phenotype of *rad52Δ* cells (*Figure 1c*), as indicated also by the six-fold reduction in survival at 200Gy compared with WT (*Figure 1e*). Nevertheless, the survival rate of *rad52-Y376A* cells was higher than that of Rad52-deficient cells (*Figure 1c*). Similarly, we found that the survival rate of cells in which the *YEKF* motif was deleted was much higher than that of Rad52-deficient cells, independently of the strain background (*Figure 1f*). The relatively lower γ-ray and MMS sensitivity of cells bearing mutations around the YEKFAP-binding domain compared with Rad52-deficient cells suggests that this interaction is largely dispensable for the formation of functional Rad51 filaments. This hypothesis was strengthened by the finding that inactivation of the *SRS2* gene fully suppressed the γ-ray and MMS sensitivity of *rad52-Y376A* cells (*Figure 1c,e*). Therefore, the sensitivity of *rad52-Y376A* cells to genotoxic agents is not related to a reduced Rad52 mediator activity, but results from Srs2 activity. This suggests a completely new view on Rad51 filament homeostasis because Rad52 appears to be essential for Rad51 filament assembly only through its interaction with RPA, whereas its interaction with Rad51 is dispensable. The Rad51-Rad52 interaction seems to be mostly required for the protection of Rad51 filaments from disruption by the translocase Srs2, and for the toxicity of Rad51 filaments in Srs2-deficient cells.

## The *rad52-Y376A* and *rad52-P381S* mutations impact the interaction between Rad52 and Rad51

To further address the relationship between the potential toxicity of Rad51 filaments and the interaction of Rad51 with Rad52, we tested the effect of Rad52 mutations around the YEKFAP domain on the capacity of FLAG-tagged Rad52 to co-immunoprecipitate with Rad51 (*Figure 2a*). First, we confirmed that the interaction between Rad52 and Rad51 cannot be detected in cells that carry a deletion in the Rad52-YEKF motif (*Krejci et al., 2002*). We obtained the same result in *rad52-Y376A*

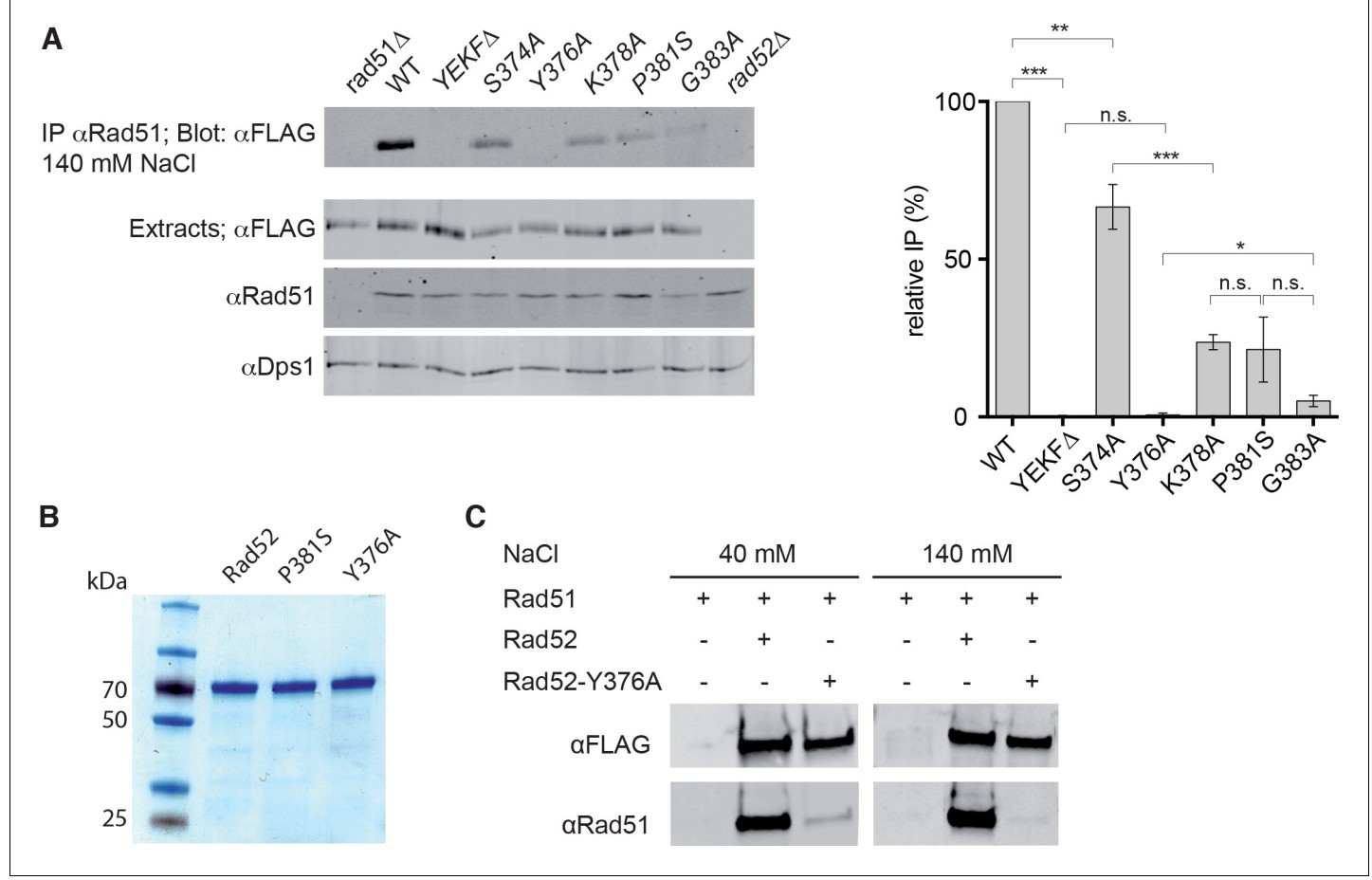

**Figure 2.** Measurement of Rad52-Rad51 interaction in cells that express various *Rad52-FLAG* mutants. (**A**) Rad51 immunoprecipitation from cells that express different Rad52-FLAG mutants. The anti-Dps1 antibody was used as loading control. The signals corresponding to immunoprecipitated Rad52 mutants (Blot: αFLAG) were quantified relatively to the signal for WT Rad52, corrected to the quantity of Rad52 in each extracts (Extracts: αFLAG) and to the Dps1 signal (αDps1). Data are presented as the mean ± SEM (n = 3). Fisher's exact test, n.s. p>0.05, *p<0.05, **p<0.01, ***p<0.001. (**B**) Purification of recombinant Rad52-FLAG, Rad52-P381S-FLAG and Rad52-Y376A-FLAG. (**C**) Pull-down experiments using Rad52-FLAG WT or Y376A fusion protein in the presence of different salt concentrations.

DOI: https://doi.org/10.7554/eLife.32744.006

The following source data is available for figure 2:

**Source data 1.** Relative IP (%) and Two-tailed p values, unpaired T test for *Figure 2A*.

DOI: https://doi.org/10.7554/eLife.32744.007

cells, whereas in *rad52-P381S*, *rad52-K378A* and *rad52-G383A* cells the interaction was detectable but significantly reduced compared with WT. Conversely, in *rad52-S374A* cells in which the growth defect of *srs2Δ* cells on MMS-containing medium is not suppressed (*Figure 1—figure supplement 1b*), the Rad52-Rad51 interaction was only slightly affected. Altogether, these results indicate that there is a correlation between the ability of the *rad52* mutants to interact with Rad51 and their capacity to suppress *srs2Δ* cells phenotypes. This suggests a correlation between the Rad52-Rad51 interaction and the potential toxicity of Rad51 filaments.

To verify that the mutations in the binding domain have a direct effect on the interaction between Rad52 and Rad51, we measured in vitro the consequences of the mutation that affected the most the interaction (i.e. *rad52-Y376A*). Pull-down experiments with purified Rad52 and Rad52-Y376A FLAG-tagged proteins (*Figure 2b*) showed that this mutation impaired very strongly the interaction with Rad51 (*Figure 2c*). By performing pull-down assays in the presence of low-salt concentrations (40 mM instead of 140 mM NaCl), we observed a residual interaction between Rad51 and Rad52-

Y376A-FLAG. We cannot fully exclude that this residual interaction provides mediator activity to Rad52-Y376A-FLAG.

## Rad52 interaction with Rad51 is dispensable for gene conversion

To strengthen our conclusion that the interaction between Rad52 and Rad51 is not required for Rad51 filament formation, we measured the rate of γ-ray-induced HR between *arg4-RV* and *arg4-Bg* heteroalleles in diploid cells (*Figure 3a*). We decided to focus on the *rad52-P381S* and *rad52-Y376A* mutations because they are representative of a partial and a strong loss of interaction, respectively. Like in haploid cells, *rad52-P381S* homozygous diploids were only slightly more sensitive to γ-ray irradiation than WT cells, whereas *rad52-Y376A* showed much higher sensitivity (*Figure 3b*). However, HR frequencies were comparable in mutant and WT cells, indicating that the impaired interaction between Rad52 and Rad51 does not prevent the recovery of ARG$^+$ recombinants in the surviving cells.

Remarkably, the strong hyper-recombination phenotype associated with acute sensitivity to γ-ray of Srs2-deficient diploid cells was partially suppressed by the *rad52-P381S* mutation and strongly suppressed by *rad52-Y376A*. Indeed, *rad52-Y376A srs2Δ* homozygous diploid cells were 1000 times more resistant than *srs2Δ* cells to exposure to 400 Gy and the HR frequency was 800-fold reduced (*Figure 3c*). This indicates that the Rad52-Rad51 interaction is responsible for the high lethality and strong hyper-recombination phenotype observed in Srs2-deficient cells.

We also measured the effect of both *RAD52* mutants on DSB repair. To this aim, we used a genetic system that allows the repair of a HO-induced DSB at the *MAT* locus of haploid cells (chromosome III) by gene conversion using another *MAT* copy located on chromosome V (*Figure 3d*) (*Ira et al., 2003*). We observed that the *rad52-P381S* and *rad52-Y376A* mutations affected cell survival only marginally, again indicating that the Rad52-Rad51 interaction is not essential for DSB repair by HR, and by extension for Rad51 filament formation. As observed before (*Ira et al., 2003*), DSB formation strongly reduced survival of Srs2-deficient cells. These authors showed that cell lethality following DSB formation was associated with increased crossing-over rate. This was interpreted as a failure to resolve HR events through the synthesis-dependent strand annealing (SDSA) pathway (gene conversion without crossing-over) in the absence of Srs2 activity. The *RAD52* mutants had only a marginal effect on survival of Srs2-deficient cells, indicating that they cannot rescue the SDSA defect.

## Rad52 interaction with Rad51 is dispensable for Rad51 filament formation at a HO-induced DSB, but is essential for preventing Rad51 filament disassembly by Srs2

Our data suggest that the Rad52-Rad51 interaction is not essential for Rad51 filament formation. To test this hypothesis, we measured the recruitment of proteins involved in Rad51 filament formation by chromatin immunoprecipitation experiments (ChIP) in haploid cells that express WT or mutant Rad52-FLAG. We used a system designed by *Vaze et al. (2002)* where a HO-induced DSB can be repaired by single strand annealing (SSA) between direct repeats located 25 kb apart (SSA assay) (*Figure 4a*). This assay involves the formation of long 3'-end ssDNA tails generated from the DSB, thus ensuring the sensitive detection of RPA, Rad52-FLAG and Rad51 recruitment to the DSB site. Quantitative PCR assays using primer sets that amplify DNA sequences at 0.6 or 7.6 kb upstream of the DSB site at different time points after DSB induction (*Figure 4b*) showed an increase of the relative enrichment of RPA, Rad52-FLAG and Rad51 at the site of DSB formation compared with the uncut *ARG5,6* locus. RPA, Rad52-FLAG and Rad51 loading peaked after 4 hr and was lower at 7.6 kb (green) than at 0.6 kb (red) (*Figure 4b*), as described previously (*Esta et al., 2013*). This is indicative of RPA, Rad52-FLAG and Rad51 loading after ssDNA formation. As previously described, Rad51 enrichment increased in *srs2Δ* cells (*Esta et al., 2013*), particularly at 7.6 kb, showing that Srs2 displaces Rad51 much more efficiently at distant sites.

Compared with WT cells, Rad51 loading did not significantly change in cells that express Rad52-P381S-FLAG (*Figure 4—figure supplement 1b*), whereas it was significantly reduced by 2.2-fold in *rad52-Y376A-FLAG* cells (*Figure 4b*). Conversely, RPA loading was 1.5-fold increased. This could be interpreted as a deficiency of Rad52 mediator activity. However, in *rad52-Y376A-FLAG srs2Δ* cells, Rad51 loading was 3.6-fold higher than in *rad52-Y376A-FLAG* cells and similar to that in *srs2Δ* cells

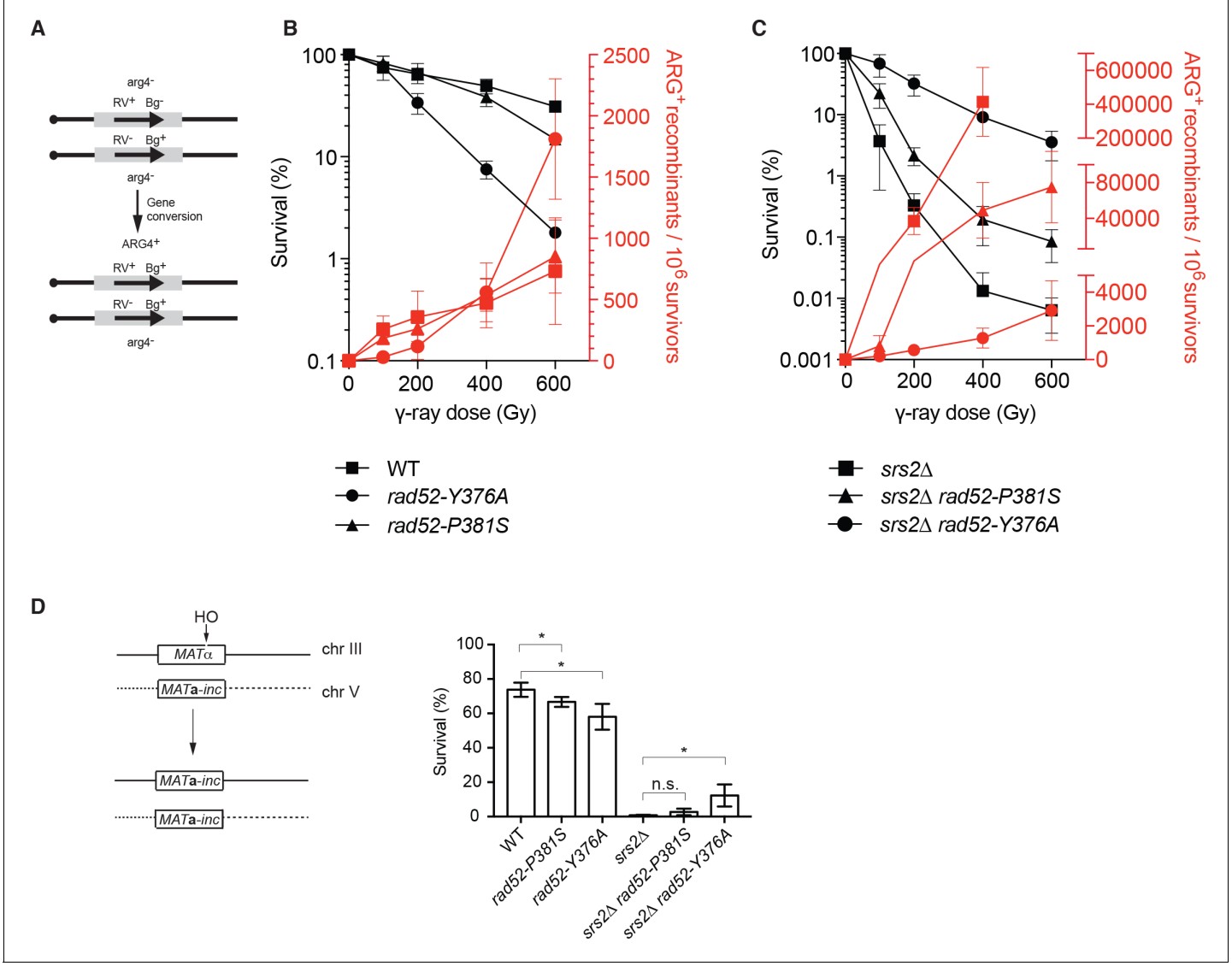

**Figure 3.** Effect of the *rad52-P381S* and *rad52-Y376A* alleles on HR. (**A**) Recombination system allowing the formation of a WT *ARG4* allele by gene conversion of the *arg4-RV* or *arg4-Bg* heteroalleles in diploid cells. (**B, C**) Survival curves (black) and heteroallelic HR frequencies (red) for the indicated homozygous diploid cells exposed to γ-rays. Data are presented as the mean ± SEM of at least three independent experiments. (**D**) HO-induced gene conversion between *MAT* ectopic copies. Cell viability after DSB formation is shown as the mean ± SEM of at least three independent experiments. Fisher's exact test, n.s. p>0.05, *p<0.05.

DOI: https://doi.org/10.7554/eLife.32744.008
The following source data is available for figure 3:

**Source data 1.** Source data for *Figure 3B,C and D*.
DOI: https://doi.org/10.7554/eLife.32744.009

(*Figure 4b*). Therefore, the reduction in Rad51 loading observed in *rad52-Y376A-FLAG* cells fully depended on Srs2 activity. These results confirmed that the Rad52-Rad51 interaction is not required to recruit Rad51 at the DSB site, and that the association between Rad52 and Rad51 is crucial for preventing Rad51 filament disassembly by Srs2.

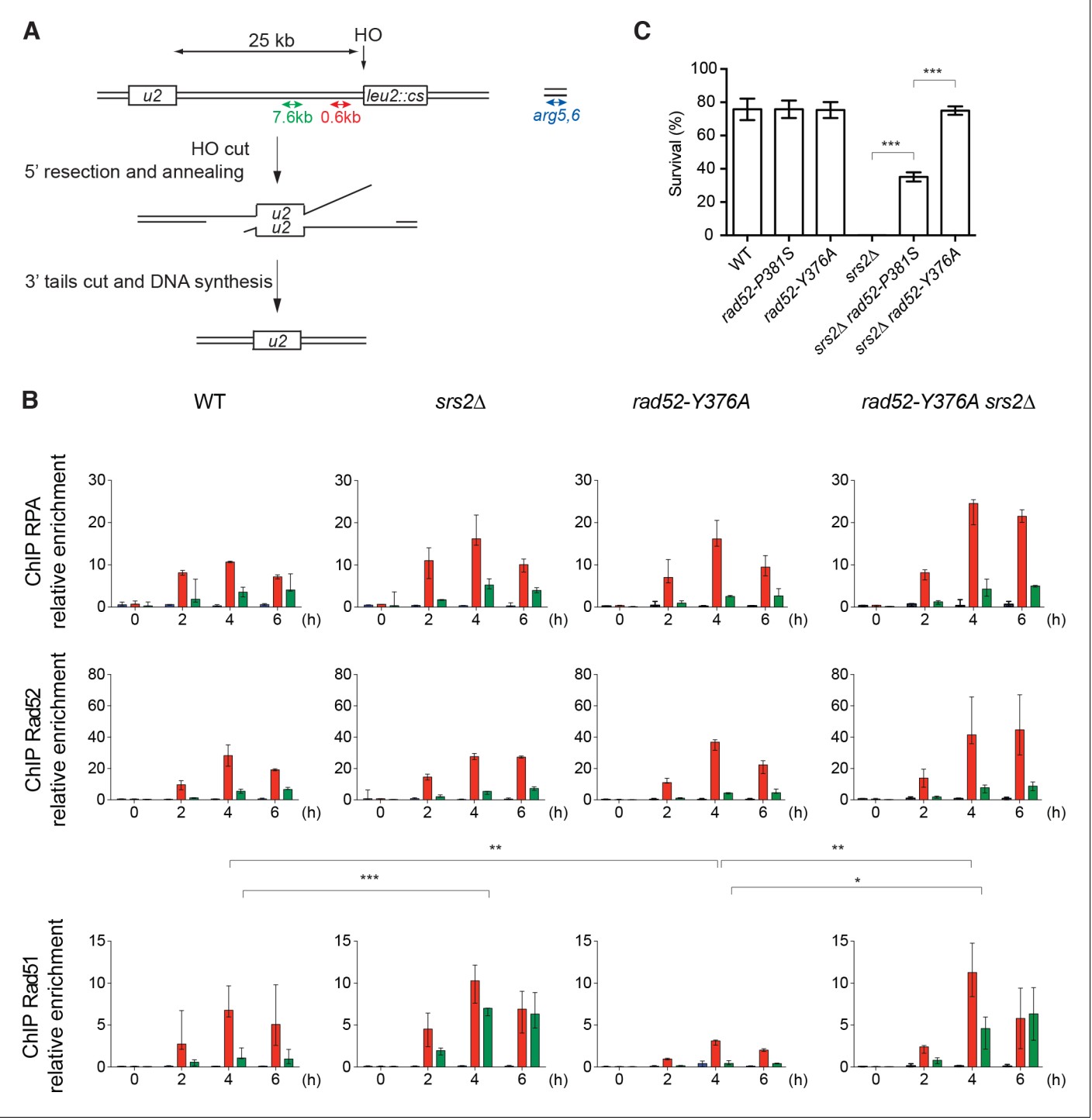

**Figure 4.** ChIP analysis of Rad51 filament formation at a DSB created by the HO endonuclease. (**A**) Schematic of the HO-induced SSA repair system. (**B**) ChIP was used to assess RPA, Rad52 and Rad51 relative enrichment at 0.6 or 7.6 kb from the DSB site and at the uncut *ARG5,6* locus at the indicated time points (hours) after HO induction. The median value (n ≥ 2) is shown and error bars represent the upper and lower values measured. Fisher's exact test, *p<0.05, **p<0.01, ***p<0.001. Statistical analysis performed 4 hr after HO-induction; only significant differences are shown. (**C**) Cell survival after HO-induced DSB formation. Cell viability after DSB formation is shown as the mean ± SEM of at least three independent experiments. Fisher's exact test, ***p<0.001.

DOI: https://doi.org/10.7554/eLife.32744.010

The following source data and figure supplements are available for figure 4:

**Source data 1.** Source data for *Figure 4A*.

*Figure 4 continued on next page*

*Figure 4 continued*

DOI: https://doi.org/10.7554/eLife.32744.013

**Figure supplement 1.** ChIP analysis of Rad51 filament formation in *rad52-P381S* mutant cells that lack or not Srs2.

DOI: https://doi.org/10.7554/eLife.32744.011

**Figure supplement 1—source data 1.** Source data for *Figure 4—figure supplement 1*.

DOI: https://doi.org/10.7554/eLife.32744.012

## The Rad52-Rad51 interaction is required for the toxicity of Rad51 filaments forming around a DSB in Srs2-deficient cells

Cell survival after DSB formation in the SSA assay depends on *RAD52* (*Vaze et al., 2002*), probably through its ssDNA pairing activity that catalyzes the annealing between homologous ssDNA. However, the *rad52-P381S* and *rad52-Y376A* mutations did not affect cell survival (*Figure 4c*). This suggested that these Rad52 mutations do not alter Rad52 pairing activity. Srs2 also is essential for cell survival, a result confirmed here (*Figure 4c*), but not Rad51 (*Vaze et al., 2002*). It has been proposed that Srs2 is required to remove Rad51 that accumulates on long 3'-ssDNA generated from DSB processing (*Vasianovich et al., 2017*) and to avoid the formation of branched toxic joint molecules upon ssDNA invasion at ectopic positions (*Elango et al., 2017*). On the other hand, viability was partially improved in *rad52-P381S srs2Δ* cells and fully restored in *rad52-Y376A srs2Δ* cells (*Figure 4c*). As our ChIP experiments showed that Rad51 is recruited normally in these double mutants, we think that this suppressor phenotype is not the consequence of impaired Rad51 filament formation. We propose instead that losing the interaction between Rad52 and Rad51 leads to less stable Rad51 filaments that are less toxic on long 3'-ssDNA in the absence of Srs2 activity.

## In vitro analysis of Rad52 activities when interaction with Rad51 is altered

Altogether, our results suggested that the Rad52-Rad51 interaction is not essential for Rad51 filament formation, but makes Rad51 filaments toxic in Srs2-deficient cells. In Srs2-proficient cells, Rad52-Rad51 interaction could be essential to protect Rad51 filaments from Srs2 dismantling activity. To further test these hypotheses, we examined the effect of mutations that affect Rad52-Rad51 interaction on Rad52 biochemical properties in vitro. Using electrophoretic mobility shift assays, we found that the mutated proteins Rad52-P381S and Rad52-Y376A (FLAG-tagged) bound to ssDNA and dsDNA similarly to WT (*Figure 5—figure supplement 1a*). Moreover, they annealed complementary ssDNA strands as efficiently as WT and were subject to inhibition by RPA and Rad51 as well (*Figure 5—figure supplement 1b*; *Wu et al., 2008*).

We then used electron microscopy (EM) to determine the effect of Rad52 mutations on Rad51 filament formation on 5 kb-long φX174 viral (+) ssDNA (*Figure 5a*). As described before (reviewed in *San Filippo et al., 2008*), incubation of Rad51 with φX174 viral (+) ssDNA prior to RPA addition produced complete Rad51 filaments, on which Rad51 fully covered ssDNA. Conversely, addition of Rad51 to RPA-coated ssDNA mostly led to the production of partial Rad51 filaments. The concomitant addition of Rad52 and Rad51 to RPA-coated ssDNA allowed the formation of complete Rad51 filaments (100%), confirming that Rad52 overcomes RPA inhibitory effect. Addition of Rad52-P381S or Rad52-Y376A, instead of Rad52, to RPA-coated ssDNA led mostly to the formation of complete Rad51 filaments (85 and 73%, respectively). However, the significant increase in the amount of RPA filaments still present at the end of the reaction suggests a slight defect in the capacity of these Rad52 mutants to start the nucleation process that initiates Rad51 filament formation.

These results confirmed that despite their weak ability to interact with Rad51, the mutant proteins can bypass RPA and form Rad51 filaments with high efficiency. They also confirmed our previous observation (*Esta et al., 2013*) that Rad52 remains associated with Rad51 filaments (*Figure 5a*). The two Rad52 mutant proteins also remained associated with Rad51 filaments, albeit to a lower extent compared with WT Rad52 (*Figure 5b*). This was surprising because the Rad52-Y376A variant harbors a mutation that largely impairs the interaction with Rad51 (our co-immunoprecipitation and pull-down experiments; *Figure 2a,c*). This difference could be explained by the low-salt concentration classically used for Rad51 filament formation in EM experiments (40 mM KCl). Indeed, Rad51

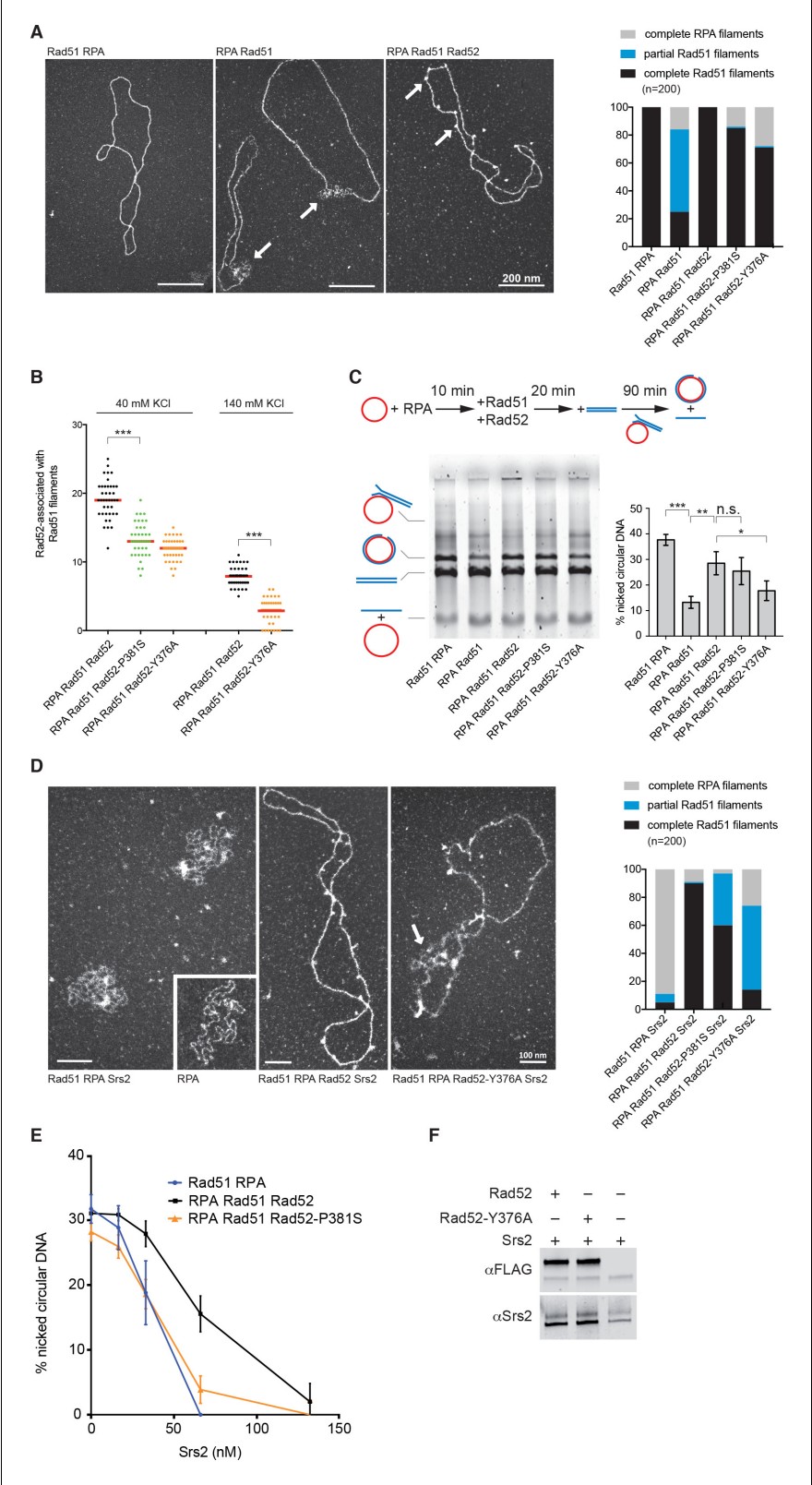

**Figure 5.** Biochemical analysis of Rad52-P381S and Rad52-Y376A mutant proteins. All experiments were performed with FLAG-tagged Rad52 proteins. (**A**) EM analysis of Rad51 filament formation. A representative nucleofilament issued from each reaction is shown. Addition of Rad51 prior to RPA on circular φx174 ssDNA (Rad51 RPA) forms complete Rad51 filaments, whereas addition of RPA before Rad51 (RPA Rad51) results in partial Rad51 filaments. White arrows indicate the presence of RPA on ssDNA. Concomitant addition of Rad52 with Rad51 in this reaction (RPA Rad51 Rad52) leads to

*Figure 5 continued on next page*

*Figure 5 continued*

complete Rad51 filaments associated with Rad52 (white arrows). The percentage of each molecular species is plotted on the histogram (right panel), showing the weak effect of Rad52 mutants on Rad51 filament formation. Two independent experiments were performed with very similar results. The results of the individual biological replicates were pooled (300 molecules analyzed). (B) Number of Rad52, Rad52-P381S and Rad52-Y376A molecules associated with Rad51 filaments assembled at two NaCl concentrations (n = 40). The median value is represented by the horizontal bar. Fisher's exact test, ***p<0.001. (C) Effect of Rad52 mutants on strand exchange. The standard reaction (Rad51 RPA) shows that complete Rad51 filaments efficiently catalyze the formation of nicked circular products. Pre-bound RPA inhibits this reaction (RPA Rad51); however, the inhibitory effect of pre-bound RPA is overcome by Rad52 (RPA Rad51 Rad52) and, to a lesser extent, by the Rad52 mutants tested. The percentage of nicked circular product calculated by quantification of the gels is shown. Data are presented as the mean ± SEM of at least three independent experiments. Fisher's exact test, n.s. p>0.05, *p<0.05, **p<0.01, ***p<0.001. (D) EM analysis of Rad51 filament displacement by Srs2. Srs2 was added for 5 min at the end of the Rad51 filament formation reactions (Rad51 RPA Srs2 and RPA Rad51 Rad52 Srs2). The white arrow shows the appearance of RPA on Rad52-Y376A-assembled Rad51 filaments after Srs2 addition. The percentage of each molecular species is shown in the histogram. Two independent experiments were performed with very similar results. The results of the individual biological replicates were pooled (200 molecules analyzed). (E) Rad52 increases strand exchange efficiency in the presence of Srs2. Strand exchange reactions were performed as in (C) except that increasing concentrations of Srs2 were added with the linear dsDNA for 90 min. Data are shown as the mean ± SEM of at least three independent experiments. (F) Pull-down assay with an anti-FLAG antibody shows the interaction between Rad52 and Srs2.

DOI: https://doi.org/10.7554/eLife.32744.014

The following source data and figure supplements are available for figure 5:

**Source data 1.** Source data for *Figure 5E*.
DOI: https://doi.org/10.7554/eLife.32744.020
**Figure supplement 1.** DNA binding and strand annealing activity of Rad52 mutants.
DOI: https://doi.org/10.7554/eLife.32744.015
**Figure supplement 1—source data 1.** DNA annealed (%) for *Figure 5—figure supplement 1B*.
DOI: https://doi.org/10.7554/eLife.32744.016
**Figure supplement 2.** Rad51 filament stabilization by Rad52.
DOI: https://doi.org/10.7554/eLife.32744.017
**Figure supplement 3.** Measurement of Srs2 ATPase activity.
DOI: https://doi.org/10.7554/eLife.32744.018
**Figure supplement 3—source data 1.** ATP hydrolysis (%) for *Figure 5—figure supplement 3*.
DOI: https://doi.org/10.7554/eLife.32744.019

filament formation in the presence of 140 mM KCl led to a weaker association of WT and mutant Rad52 with Rad51 filaments, but Rad52-Y376A was still associated with Rad51 filaments.

Previous experiments showed that the in vitro φX174-based strand exchange reaction is significantly affected by *rad52-YEKFΔ* (*Krejci et al., 2002*). However, we found that Rad52-P381S and Rad52-Y376A did not strongly affect Rad51 filament formation on φX174 ssDNA and that they efficiently promoted HR in vivo. Therefore, we tested the two mutated proteins in the strand exchange experiment (*Figure 5c*). As previously described, RPA pre-bound to ssDNA reduced strand exchange by approximately fourfold compared to the standard reaction, where Rad51 is added prior to RPA. The concomitant addition of Rad52 and Rad51 released the inhibition by pre-bound RPA and led to a threefold increase in strand exchange. We found that Rad52-P381S can release the RPA inhibition as efficiently as the WT protein, whereas Rad52-Y376A is affected, confirming the results obtained with Rad52-YEKFΔ. Although Rad51 filament formation mediated by the mutated Rad52 proteins was only slightly affected, strand exchange in vitro was strongly affected. This is in contrast with our observation in vivo, suggesting that other activities could compensate for the Rad52 mutations in vivo.

Finally, we tested whether Rad52-Rad51 interaction stabilizes Rad51 filaments. This effect could explain how Rad52 triggers Rad51 filament toxicity in Srs2-deficient cells. We previously demonstrated that the association of Rad52 with Rad51 filaments is challenged at salt concentrations lower than those required to destabilize Rad51 filaments (*Esta et al., 2013*). Therefore, we challenged the stability of Rad51 filaments by incubation with excess ssDNA (ΦX174 viral (+) strand). This experiment could not be done by EM, because the competing DNA hinders the spreading of Rad51 filaments on carbon grids. Rad51 filaments were formed by adding Rad51 with Rad52 or Rad52-Y376A on a 400nucleotides (nt)-long Cy5-labeled ssDNA pre-coated with RPA. The previously described optimal stoichiometric conditions were used with slight modifications (*Esta et al., 2013*; see Materials and methods). Analysis of glutaraldehyde-fixed-protein complexes by agarose gel

electrophoresis (*Figure 5—figure supplement 2a*) showed that Rad52-Y376A assembled Rad51 filaments as efficiently as WT Rad52. This confirmed our EM observation and our in vivo results that the Rad52-Rad51 interaction is not strictly required for Rad51 filament formation. After incubation at 37°C for 20 min to allow Rad51 filament formation, competing ΦX174 viral (+) strand was added to the reaction and incubated for an additional 30 min. Surprisingly, the stability of Rad51 filaments assembled with Rad52 or Rad52-Y376A was comparable (*Figure 5—figure supplement 2a*). We confirmed this result using a benzonase assay (*Figure 5—figure supplement 2b*). This suggested that losing the interaction between Rad52 and Rad51 does not affect the stability of Rad51 filaments. However, we previously showed by EM that only 10% of Rad51 filaments assembled on the same 400nt-long ssDNA are associated with more than one Rad52 spot (*Esta et al., 2013*). Therefore, we cannot rule out that the amount of Rad52 associated with Rad51 filaments was too low to show an effect on stability in this assay and we cannot conclude on the effect of Rad52 on Rad51 filament stability.

## Rad52 protects Rad51 filaments from Srs2 dismantling activity in vitro

To test whether Rad52 protects Rad51 filaments from Srs2 dismantling activity, we incubated Srs2 with Rad51 filaments assembled in vitro on ΦX174 ssDNA for 5 min, and analyzed the resulting nucleoprotein complexes by EM. As described previously (*Veaute et al., 2003*; *Krejci et al., 2003*), Srs2 efficiently disrupted Rad51 filaments when Rad51 was added prior to RPA. Only 11% of DNA molecules remained covered with Rad51 after 5 min of incubation with Srs2 (*Figure 5d*). Conversely, when Rad52 and Rad51 were added together on RPA-coated ssDNA, 90% of complete Rad52-associated Rad51 filaments were still present after 5 min of incubation with Srs2. This result clearly indicated that Rad52 protects Rad51 filaments from Srs2 dismantling activity. This protection was weakened when Rad51 filaments were formed in the presence of Rad52-P381S or Rad52-Y376A (only 60 and 14% of complete Rad51 filaments after 5 min).

As Srs2 also inhibits the φX174-based strand exchange reaction in vitro (*Veaute et al., 2003*), we used this assay to confirm Rad52 protective effect. Addition of 66 nM Srs2 was enough to inhibit the strand exchange reaction when Rad51 was added to ssDNA prior to RPA (*Figure 5e*). Conversely, when Rad52 was added with Rad51 on RPA-coated ssDNA, half of the products were still formed in the presence of the same concentration of Srs2. Double Srs2 concentration was required to efficiently inhibit the reaction in the presence of Rad52. Srs2-mediated inhibition was dependent on Rad52-Rad51 interaction because strand exchange catalyzed by Rad52-P381S was largely reduced.

Rad52 capacity to protect Rad51 filaments could be the consequence of a direct interaction with Srs2, and we found by pull-down assay that Rad52 and Srs2 interacted. This interaction was not affected by the *rad52-Y376A* mutation (*Figure 5f*). Moreover, Rad52 did not affect Srs2 ATPase activity (*Figure 5—figure supplement 3*). Altogether, these results suggest that the presence of Rad52 within Rad51 filaments controls Srs2, probably through a competition mechanism rather than by direct inhibition of the Rad51-catalyzed Srs2 translocase activity.

## Discussion

### Rad52 interaction with Rad51 is responsible for the toxicity of Rad51 filaments in Srs2-deficient cells

We previously showed that Rad52 is involved in the toxicity of Rad51 filaments in Srs2-deficient cells (*Esta et al., 2013*). To understand the underlying mechanism, we conducted a *RAD52* gene mutation screen to identify mutations that could suppress the toxicity of Rad51 filaments without affecting their formation. Among the hits, we found several mutations in amino acid residues located within the Rad52 domain that contains an essential motif for its interaction with Rad51 (*Krejci et al., 2002*). The extended analysis of this domain by directed mutagenesis showed that there is a correlation between the impairment of the Rad52-Rad51 interaction by these mutations and their ability to suppress the Srs2-deficient cell phenotype. Specifically, *rad52-Y376A* restores resistance to DNA damage in *srs2Δ* homozygous diploid cells exposed to γ-rays as well as viability of *srs2Δ* haploid cells in which a HO-induced DSB is repaired by SSA. The *rad52-P381S* mutation, which affects the Rad52-Rad51 interaction less strongly than *rad52-Y376A*, is a weaker suppressor of the Srs2-deficient cells

phenotype. The *rad52-Y376A* mutation also suppresses synthetic lethality induced in Srs2-deficient cells by mutations in genes involved in HR or in DNA replication. ChIP analyses confirmed that the impaired interaction between Rad52 and Rad51 does not affect Rad51 recruitment to ssDNA at DSB sites in Srs2-deficient cells. Thus, from these in vivo observations we infer that the suppression of the *srs2Δ* cell phenotype cannot be attributed to a lower mediator activity of Rad52-Y376A. In turn, this suggests that the potential toxicity of Rad51 filaments in Srs2-deficient cells is related to the association of Rad52 with Rad51 filaments. In agreement, our EM analysis of Rad51 filament assembly on φX174 ssDNA indicates that the association of Rad52-Y376A with Rad51 filaments is reduced compared with WT Rad52.

## How does Rad52 association with Rad51 filaments lead to toxicity in the absence of Srs2?

Surprisingly, we found that the *rad52-Y376A* mutation strongly impairs the interaction with Rad51 (co-immunoprecipitation and pull-down experiments), while it affects only partially the association of Rad52 with Rad51 filaments observed by EM. We hypothesize that this mutation affects mostly the interaction between protomers, which could be the most common interaction in the nucleus. On the other hand, the interaction with Rad51 filaments could be complex. Besides its interaction with Rad51 through the YEKFAP domain, Rad52 could associates with Rad51 filaments through DNA binding (*Mortensen et al., 1996*), or via interaction with residual DNA-bound RPA. Moreover, the Rad52 FVTA motif, which is essential for binding to Rad51 (*Kagawa et al., 2014*), could also be involved in the association of Rad52 with Rad51 nucleofilaments.

Similarly to what was proposed for the Rad51 paralogues SWS-1 in *Caenorhabditis elegans* (*Taylor et al., 2016*; *McClendon et al., 2016*) and Rad55/Rad57 in yeast (*Liu et al., 2011*), we suggest that Rad52 could be involved in the stabilization of Rad51 filaments in vivo in yeast. Although we did not see any effect of the *rad52-Y376A* mutation on the stability of Rad51 filaments (ssDNA competition assay) or on protection against benzonase endonuclease activity, it is clear that this mutation suppresses Rad51 filament toxicity and rescues most of the phenotypes of Srs2-deficient haploids and diploids cells. It is possible that the toxic effect of Rad52 binding to Rad51 filaments requires the association with the yeast Rad51 paralogues Rad55/Rad57 and the SHU complex. Rad52-stabilized Rad51 nucleofilaments might lead to different disturbances that cause the toxicity observed in Srs2-deficient cells. For example, *Vasianovich et al. (2017)* recently proposed that Rad51 has to be removed from ssDNA to allow de novo DNA synthesis to fill in the gaps formed during the SSA process. Rad52-mediated stabilization of Rad51 filaments could also favor the formation of branched toxic joint molecules upon ssDNA invasion at ectopic positions (*Elango et al., 2017*).

## The Rad51-binding domain of Rad52 is dispensable for Rad51 filament assembly in vivo

Rad52-Y376A can properly assemble Rad51 filaments on a 400nt-long ssDNA and is only slightly defective on φX174 ssDNA (5 kb in size). Our EM analysis showed that 70% of complete Rad51 filaments are formed in the presence of mutant Rad52, compared with 100% with WT Rad52 and 25% in its absence. However, our experiments confirmed that disturbing the interaction between Rad52 and Rad51 strongly affects the φX174 plasmid-based strand exchange reaction (*Krejci et al., 2002*). Rad52 association with Rad51 filaments might be required for strand exchange in vitro. In contrast, Rad52-Y376A promotes strand invasion efficiently in vivo as supported by the normal frequency of γ-ray- and HO-induced HR in *rad52-Y376A* mutant cells. Additionally, the weak sensitivity of *rad52-Y376A* cells to γ-rays and the lower recruitment of Rad51 on ssDNA after DSB formation in *rad52-Y376A* cells are both dependent on Srs2. We hypothesize that the different results obtained in vivo and in vitro might also be related to a compensatory effect in vivo by the Rad51 paralogues Rad55/Rad57 and SHU.

## Rad52 protects Rad51 filaments from Srs2 dismantling activity

The fact that the sensitivity of *rad52-Y376A* mutant cells to DNA damage depends on Srs2 suggests that Rad52 protects Rad51 filaments from Srs2 activity, a hypothesis that we confirmed in vitro. EM analysis and strand exchange reactions showed that Rad51 filaments are more resistant

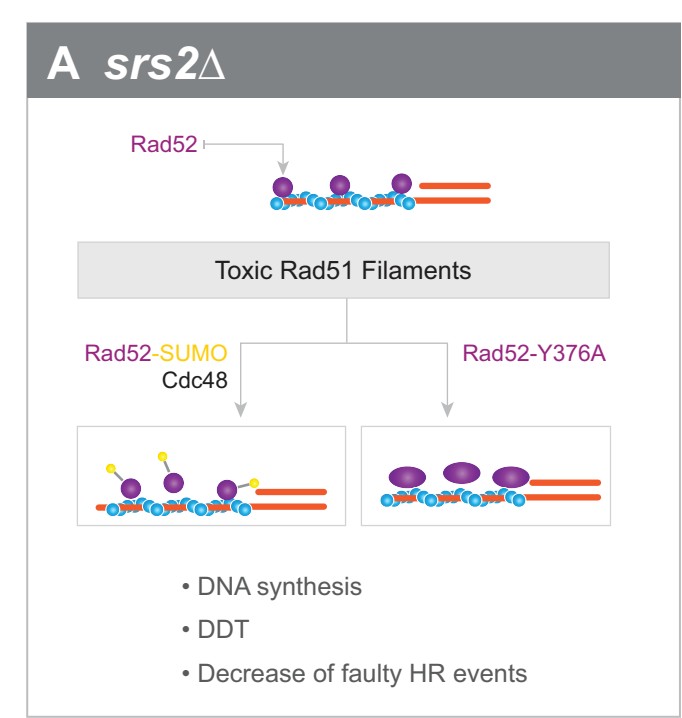

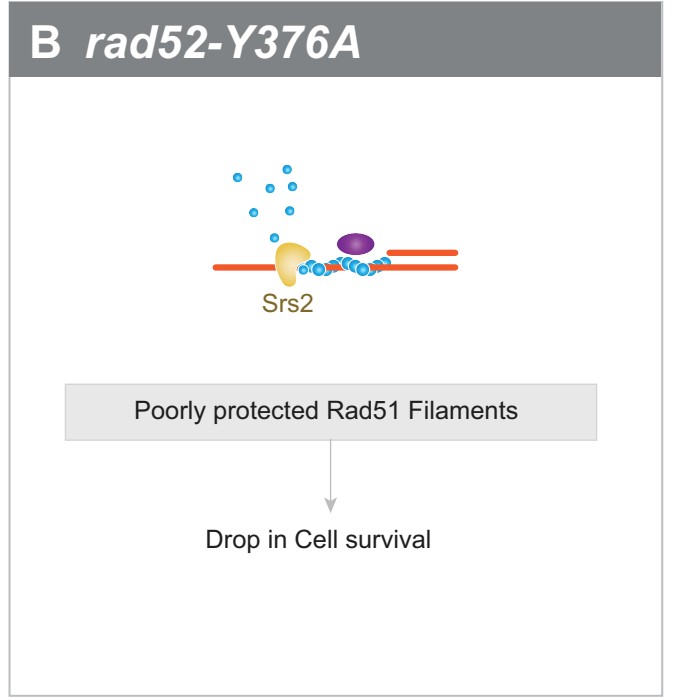

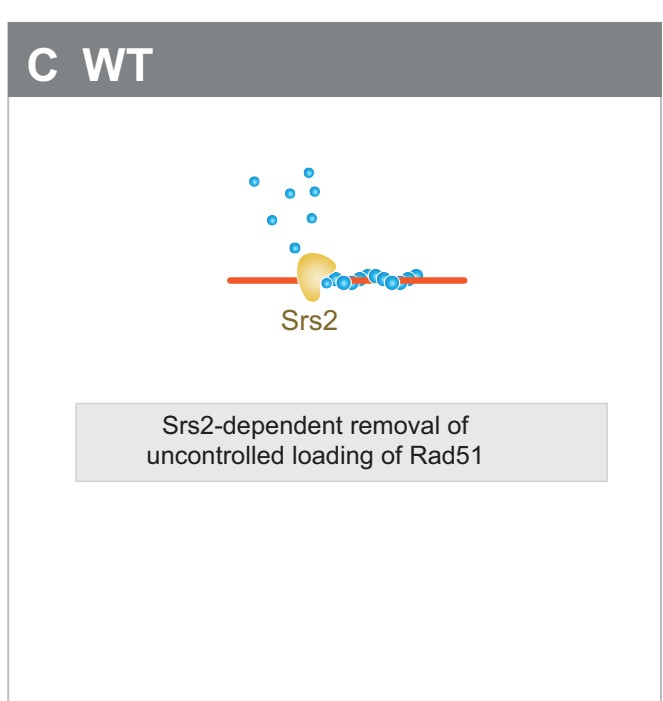

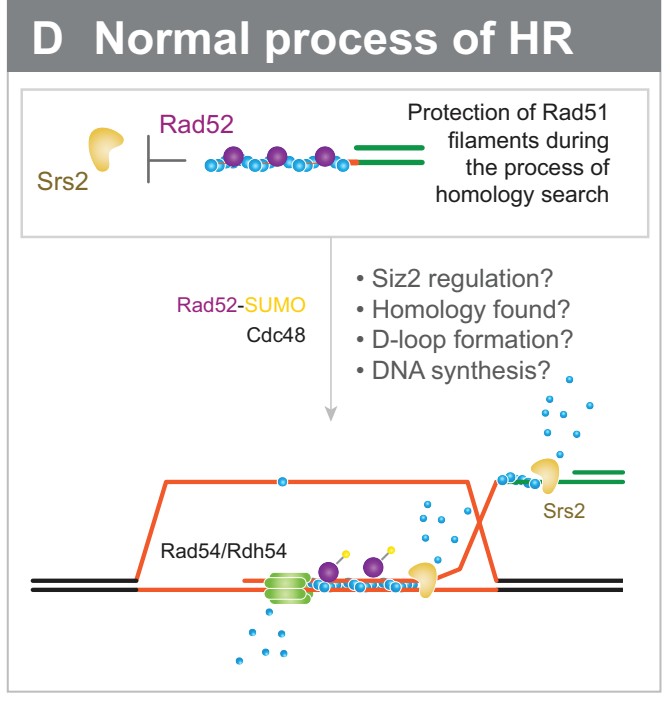

**Figure 6.** Interplay between Rad52 sumoylation and Srs2 in the control of Rad51 filament formation. (**A**) In Srs2-deficient cells, the association of Rad52 with Rad51 filaments is responsible for their potential toxicity. Massive sumoylation of Rad52 could trigger the Cdc48-dependent release of Rad52, as do mutations that affect the interaction between Rad52 and Rad51. Excessive stabilization of Rad51 filaments by Rad52 could block DNA synthesis or DNA damage tolerance (DTT) events and promote faulty HR events. (**B**) Impaired interaction between Rad52 and Rad51 would sensitize Rad51 filaments to Srs2 activity. (**C**) In WT cells, uncontrolled loading of Rad51 might lead to the production of toxic Rad51 filaments that are efficiently removed by Srs2. (**D**) Global view of Rad51 filament regulation during HR (see text for more details).

DOI: https://doi.org/10.7554/eLife.32744.021

to Srs2 activity in the presence of Rad52. We hypothesized that Rad52 association with Rad51 filaments could mask the Rad51 interaction sites that are essential for Srs2 dismantling activity (*Seong et al., 2009*). In agreement, excess of Rad52 can disrupt the Rad51-Srs2 interaction (*Seong et al., 2009*). The physical interaction we found between Rad52 and Srs2 might have a synergic effect on Rad51 filament protection. Although we found that Rad52-associated Rad51 filaments do not lower Srs2 ATPase activity, Srs2 might idle when it interacts with Rad52 without stopping its ATPase activity.

It has been proposed that Rad55/Rad57 and the SHU complex protect Rad51 filaments from Srs2 activity (*Bernstein et al., 2011*; *Liu et al., 2011*). As they act as a functional ensemble with Rad52 (*Gaines et al., 2015*), it is possible that each member of such complex plays its part in Rad51 filament protection.

## Interplay between Rad52 sumoylation and Srs2 in the control of Rad51 filament formation

We previously showed that *SIZ2* overexpression drives massive sumoylation of Rad52 and leads to the suppression of Rad51 filament toxicity in Srs2-deficient cells (*Esta et al., 2013*). Here, we found that impaired interaction between Rad52 and Rad51 has the same consequences. The finding that Cdc48 curbs the interaction between Rad52-SUMO and Rad51 (*Bergink et al., 2013*) links these two observations. We propose that massive sumoylation activates Cdc48. This, in turn, disengages Rad52 from Rad51 filaments, thus suppressing their potential toxicity in Srs2-deficient cells (*Figure 6a*). In WT cells, we suggest that destabilization of Rad51 filaments following Rad52 sumoylation is not fast enough and Srs2 is required to catalyze rapid Rad51 removal once Rad52 is dissociated to avoid the formation of toxic Rad51 filaments (*Figure 6b,c*). Massive sumoylation of Rad52 or enhanced dissociation of Rad52 by mutations might compensate for the slow kinetics of Rad51 destabilization in the absence of Srs2 activity.

Our findings and these hypotheses lead us to propose a global model on the involvement of Rad52 and Srs2 in HR (*Figure 6d*). Rad52 would be associated with Rad51 filaments long enough to ensure their protection against Srs2 and the stability required to promote homology search. Once this process is achieved, events, such as joint formation, D-loop formation or DNA synthesis, would trigger Siz2-dependent Rad52 sumoylation. Cdc48 would then induce Rad52 dissociation, allowing Srs2 to remove Rad51 from ssDNA. Srs2 activity would be essential to allow subsequent steps to occur, such as DNA synthesis, or to displace Rad51 to avoid improper dsDNA invasion. Srs2 might also be required to displace the invading strand and to promote synthesis-dependent strand annealing (*Dupaigne et al., 2008*; *Liu et al., 2017*). Rad52 release might also be important for Rad51 removal from dsDNA by the Rad54 and Rdh54 motor proteins (*Ceballos and Heyer, 2011*; *Mason et al., 2015*).

## Role of Rad52 in mammalian cells

While *RAD52* is essential for all homology-dependent repair in budding yeast, its role in mammalian cells has remained mysterious. Indeed, mice lacking RAD52 have a mild phenotype (for a review: *Liu and Heyer, 2011*), but RAD52 is required for the viability of BRCA2-deficient cells, the canonical RAD51 loader in human cells (*Feng et al., 2011*). Recent studies revealed a role for RAD52 pairing activity in promoting DNA synthesis following replication stress (*Bhowmick et al., 2016*; *Sotiriou et al., 2016*). This role is independent from those of BRCA2 and RAD51. However, RAD52 capacity to interact with RPA and RAD51 suggests a role in RAD51 filament formation (*Park et al., 1996*; *Shen et al., 1996*). Monitoring of RAD51 in U2OS cells depleted for the Cdc48 orthologue p97 and treated with zeocin shows a striking abnormal nuclear assembly of RAD51. Interestingly, these structures are strongly reduced by simultaneous depletion of RAD52, but barely by depletion of BRCA2. This suggests that BRCA2 is not primarily involved in RAD51 loading at these structures, but rather RAD52, like in yeast cells (*Bergink et al., 2013*). Similarly to the situation in yeast, Cdc48/p97 might be required to unload RAD52 that would otherwise stabilize these abnormal RAD51 structures. Therefore, like in yeast, mammalian RAD52 might be involved in RAD51 filament formation and stabilization in contexts that remain to be defined.

# Materials and methods

## Key resources table

| Reagent type (species) or resource | Designation | Source or reference | Identifiers | Additional information |
|---|---|---|---|---|
| Gene (*Saccharomyces cerevisiae*) | *RAD52* | NA | SGD: YML032C | |
| Gene (*S. cerevisiae*) | *RAD51* | NA | SGD: YER095W | |
| Gene (*S. cerevisiae*) | *Srs2* | NA | SGD: YJL092W | |
| Genetic reagent (*Bacteriophage φ-X174*) | viral (+) strand of φX174 DNA | | | |
| Strain, strain background (*S. cerevisiae*) | FF18733 | | | Cell line maintained in E. Coïc's lab |
| Strain, strain background (*S. cerevisiae*) | YFP17; JKM146 | | | Cell line maintained in J. Haber's lab |
| Antibody | anti-Rad51 (rabbit polyclonal) | Abcam | ab63798 | |
| Antibody | anti-FLAG (Mouse polyclonal) | Sigma | F3165 | |
| Antibody | anti-DPS1 (Rabbit polyclonal) | S. Marcand's lab | | |
| Antibody | anti-RPA (Rabbit polyclonal) | V. Géli's lab | | |
| Antibody | anti-Mouse IR800 Goat monoclonal) | Advansta | R-05061–250 | |
| Antibody | anti-Rabbit IR700 Goat monoclonal) | Advansta | R-05054–250 | |
| Antibody | anti-Mouse IR700 Goat monoclonal) | Advansta | R-05055–250 | |
| Antibody | anti-Srs2 (Goat polyclonal) | Santa-Cruz | sc-11991 | |
| Recombinant DNA reagent | YCplac111-Rad52-6His-3FLAG | This study | | progenitors: YCplac111 (GenBank: X75457, L26350) |
| Recombinant DNA reagent | Yiplac211-rad52-P381S; Yiplac211-rad52-Y376A | This study | | progenitors: Yiplac211 (GenBank: X75462, L26358) |
| Recombinant DNA reagent | pET15b-Rad52-FLAG | | | Progenitor: pET15b (addgene: 69661-3) |
| Peptide, recombinant protein | Rad52-FLAG | This study | | |
| Peptide, recombinant protein | RPA | This study | | |
| Peptide, recombinant protein | Rad51 | This study | | |
| Peptide, recombinant protein | Srs2 | This study | | |
| Peptide, recombinant protein | 3XFLAG | Sigma | F4799 | |
| Commercial assay or kit | GeneMorph II EZClone Domain Mutagenesis | Agilent Technologies | 200552–5 | |
| Commercial assay or kit | Dynabeads coupled to Protein A | Invitrogen | 11041 | |
| Commercial assay or kit | Dynabeads coupled to PanMouse | Invitrogen | 10002D | |
| Commercial assay or kit | Platinum SYBR Green qPCR SuperMix-UDG | Invitrogen | 11733–046 | |
| Software, algorithm | PSI-Blast | *Altschul et al. (1997)* | | |
| Software, algorithm | Muscle | *Edgar (2004)* | | |
| Software, algorithm | Jalview | *Waterhouse et al. (2009)* | | |
| Software, algorithm | iTEM software | Olympus, Soft Imaging Solutions | | |
| Software, algorithm | ImageQuant TL software | GE Healthcare Life Sciences | | |

### *S. cerevisiae* strains

Strains used in this study are listed in *Supplementary file 2*. Experiments were mostly conducted in the FF18733 background. Diploid cells used in survival and recombination assays were the result of crosses between isogenic haploid strains bearing the *arg4*-RV and *arg4*-Bg frame-shift mutations. The mutations *rad52-P381S* and *rad52-Y376A* were introduced in yeast cells with the pop-in pop-out technique using the integrative plasmids Yiplac211-*rad52-P381S* and Yiplac211-*rad52-Y376A*.

### Plasmids

The 6 His-3 FLAG tag was added in frame to *RAD52* in YCplac111 plasmids using the SLIC cloning method (*Li and Elledge, 2007*).

### Random mutagenesis

The Rad52 random mutation library was obtained by PCR-directed mutagenesis of a Ycplac111 centromeric plasmid that carried the *RAD52* gene (*Sacher et al., 2006*). The GeneMorph II EZClone Domain Mutagenesis method (Agilent Technologies) was used according to the manufacturer's instructions to obtain a maximum number of plasmids bearing only one mutation in the *RAD52* gene. PCR products were amplified in *Escherichia coli*. The library was extracted and transformed in the *rad52Δ srs2Δ* strains directly plated on MMS- containing YPD plates for selection.

### Directed mutagenesis

Single mutations were introduced in the desired plasmid using a PCR method adapted from *Hansson et al. (2008)*.

### Sequence alignment

Homologous sequences of *S. cerevisiae* Rad52 were retrieved using PSI-Blast searches against the nr database (*Altschul et al., 1997*; *Schäffer et al., 2001*). Multiple alignment of the full length sequences of these homologs was obtained using the Muscle software (*Edgar, 2004*). However, within the C-terminal disordered tail the algorithm did not satisfactorily align the small linear motifs that surround the Rad51 binding domain and the alignment had to be manually refined. The final alignment was represented using Jalview (*Waterhouse et al., 2009*).

### Irradiation and measurement of recombination rates

γ-ray irradiation was performed using a [137]Cs source. After irradiation, exponentially growing cells were plated at the appropriate dilution on rich medium (YPD) to measure the survival rate, and on synthetic plates without arginine to quantify the number of HR events. For quantitation, the mean percentage of survival from at least three independent experiments is presented.

### Survival following DSB formation

Cells were grown overnight in liquid culture medium containing lactate before plating. Survival following HO-induced DSB was measured as the number of cells growing on galactose-containing medium divided by the number of colonies growing on YPD. The results shown are the average of at least three independent experiments.

### Co-immunoprecipitation

Extracts were prepared from cell cultures in exponential phase as previously described (*Strahl-Bolsinger et al., 1997*) and resuspended in lysis buffer (50 mM Hepes KOH pH 7.5, 140 mM NaCl, 1 mM EDTA, 10% glycerol, 5% NP40) without DNase treatment. Whole cell extracts (1 mg) were incubated with 2 μg of a rabbit anti-Rad51 polyclonal antibody (Abcam) at 4°C for 1 hr. Then, 50 μl of Dynabeads coupled to Protein A (Invitrogen) was added, and the incubation continued for another hour. Immunoprecipitates were washed twice with 1 ml of lysis buffer and resuspended in 30 μl of 1X Laemmli buffer. Eluted proteins were analyzed by western blotting. Proteins were separated on 10% SDS-PAGE and transferred to Hybond-C Super membranes (Amersham Biosciences). Proteins were detected with a mouse anti-FLAG monoclonal (Sigma, 1/10,000), rabbit anti-Rad51 polyclonal (1/2000) or rabbit anti-DPS1 polyclonal antibody (1/30,000, a gift from S. Marcand). Blots were then incubated with monoclonal goat anti-mouse IR800 or goat anti-rabbit IR700 or IR800 secondary

antibodies (1/10,000, Advansta). Protein-antibody complexes were visualized using the Odyssey CLx system (Li-cor Biosciences). The presence of Rad51 in the immunoprecipitated fractions could not be detected to validate the efficiency of the immunoprecipitation because it migrates at the same level as the anti-Rad51 IgG. However, the absence of Rad52 in the *rad51Δ* immunoprecipitates confirmed that the detected Rad52-FLAG signals were related to the Rad52-Rad51 interaction.

## ChIP experiments and quantitative PCR analyses

Cells were grown in YPD until late exponential phase. After inoculation in 400 ml of YPLactate, cultures were grown to a concentration of 5 to $10 \times 10^6$ cells/ml. A 50 ml sample was removed at the 0 hr time-point and then galactose was added to a final concentration of 2%. Incubation was continued and 50 ml samples were removed at different time-points. Cells were fixed with 1% formaldehyde, which was subsequently neutralized with 125 mM glycine. Cells were centrifuged and washed with TBS buffer (20 mM Tris pH8, 150 mM NaCl). Cell pellets were then frozen at −20°C. ChIP was carried out as previously described with minor modifications (*Sugawara and Haber, 2006*). Samples were incubated with 2 µg of rabbit anti-RPA polyclonal (a gift from V. Géli), mouse anti-FLAG monoclonal (Sigma) or rabbit anti-Rad51 polyclonal antibody (Santa Cruz Biotechnology). 50µl of Magnetic Dynabeads Protein A or Pan Mouse (Invitrogen) was added to each sample. Following washes, protein elution and crosslink reversal, samples were incubated with proteinase K followed by DNA purification with the QIAquick PCR Purification Kit (Qiagen). The primers used for QPCR were EMO102 (CCC TGT GTG TTC TCG TTA TGT T) and EMO103 (TAA GGC GCC TGA TTC AAG A) for amplification at 0.6 kb from the DSB site, EMO65 (CCA AAT AGC CAA TGG TGT CA) and EMO66 (CCA AGA TGC AAA CCG AAT AA) for amplification at 7.6 kb from the DSB site, EMO100 (AGC AAA GTT GGG TGA AGT ATG GTA) and EMO101 (CAA ATT TGT CTA GTG TGG GAA CG) for amplification at the *ARG5,6* locus. Quantitative PCR reactions were performed using the Platinum SYBR Green qPCR SuperMix-UDG (Invitrogen) and an Eppendorf Realplex system. ChIP relative enrichment was calculated as the ratio between the quantitative PCR values of the IP fraction and those of the input fraction.

## Protein purification

Rad52-FLAG, Rad52-P381S-FLAG and Rad52-Y376A-FLAG were purified from *E. coli* BL21 (DE3) cells transformed with the respective pET15b-based plasmids. Cells were grown in 3 liters of LB broth with 100 µg/ml ampicillin at 37°C until $A_{600} = 0.8$. Protein expression was induced by addition of 1 mM IPTG followed by incubation at 30°C for 4 hr. Cells were lysed by sonication in 50 mM Tris-HCL pH 7.5, 10% sucrose, 150 mM KCl, 0.5 mM EDTA, 1 mM DTT, 1x complete protease inhibitor (Roche), 5 µg/ml chymostatin, 5 µg/ml pepstatin, 1 mM AEBSF and 0.1% NP40. Lysates were clarified by centrifugation and supernatants were precipitated with ammonium sulfate (50% saturation). Pellets were suspended in buffer Kc (20 mM $KH_2PO_4$ pH 7.4, 1X Complete protease inhibitor, 5 µg/ml chymostatin, 5 µg/ml pepstatin, 1 mM AEBSF, 500 mM KCl and 20 mM imidazole) and incubated with a saturating amount of Ni Sepharose High Performance Resin (GE Healthcare) at 4°C for 2 hr. The resin was washed with large volumes (100–150 ml) of buffer Kc. Proteins were eluted with buffer Kc containing 200 mM imidazole. The buffer was changed with a PD10 column (GE Healthcare) to buffer K (20 mM $KH_2PO_4$ pH 7.4, 1 mM DTT) containing 150 mM KCl. Next, proteins were loaded to a 1 ml Hi-trap heparin column (GE Healthcare) and eluted with a 10 ml gradient of 150–500 mM KCl in buffer K. The Rad52-containing fractions were pooled and the buffer changed to buffer K containing 120 mM KCl with a PD10 column. Proteins were loaded in a 1 ml RESOURCE Q column (GE Healthcare) and eluted through a 10 ml gradient of 120–500 mM KCl in buffer K. The Rad52-containing fractions were pooled, diluted to a final concentration of 200 mM KCl and glycerol was added (10% final concentration). Rad52 concentration was determined using an extinction coefficient of 30,495 mole/l/cm at 280 nm.

RPA was purified from the protease-deficient yeast strain BJ5496 (*ura3-52, trp1, leu2Δ1, his3Δ200, pep4::HIS3, prbΔ1.6R, can1, GAL*). Cells were transformed with three plasmids that carried the *RFA1*, *RFA2*, or *RFA3* ORF under the control of a GAL promoter (a gift from R. Kolodner). The RPA heterotrimer was purified as described (*Kantake et al., 2003*). Rad51 was overexpressed in *E. coli* BL21 (DE3) pLysS cells transformed with the pEZ5139 plasmid (provided by S. Kowalczykowski)

and then purified as described previously (*Zaitseva et al., 1999*). His-tagged Srs2 (N-terminal) was purified from sf9 cells infected with a recombinant baculovirus (*Veaute et al., 2003*).

## Pull-down assay

The interaction between Rad52 and Rad51 was assayed in vitro by incubating 80 nM Rad52-FLAG with 0.25 µg of a goat monoclonal anti-FLAG antibody in the same lysis buffer used for co-immuno-precipitation (in the presence of 40 or 140 mM NaCl), at 4°C for 30 min. Then 10 µl of Dynabeads Pan Mouse IgG (Invitrogene) were added and the incubation was continued for 1 hr. Beads were washed with lysis buffer. After addition of 5nM Rad51, the incubation was continued at 4°C for 1 hr. Samples were washed twice with lysis buffer and elution was performed by addition of 0.1 mg/ml 3XFLAG peptide (Sigma) at 30°C for 30 min. Proteins were analyzed by SDS-PAGE. Rad52-FLAG and Rad51 were detected with specific antibodies as for co-immunoprecipitation.

The interaction between Rad52 and Srs2 was assayed in vitro by incubating 4 nM Rad52-FLAG and 4 nM Srs2 with 0.5 µg anti-FLAG antibody in buffer P (25 mM Tris HCl pH 7.5, 10 mM magnesium acetate, 1 mM DTT, 90 mM NaCl, 10% glycerol, 0.05% NP40) at 25°C for 90 min. 12.5 µl of Dynabeads Pan Mouse IgG were then added to the mixture and incubated at 25°C for another 90 min. Beads were washed twice with buffer P. Pulled down proteins were eluted by boiling at 95°C in 1x Laemmli. Proteins were separated in SDS PAGE and Rad52 was revealed as described previously. Srs2 was detected with a goat anti-Srs2 polyclonal antibody (Santa Cruz, 1/2000) and with a secondary anti-goat IR800 antibody (Advansta, 1/10,000).

## Electrophoretic mobility shift assay

Increasing amounts of WT and mutant Rad52-FLAG were incubated with 0.27 µM 5' end-Cy5-labeled XV2 oligonucleotide (5'-TGG GTG AAC CTG CAG GTG GGC AAA GAT GTC CTA GCA ATG TAA TCG TCA AGC TTT ATG CCG TT-3') in buffer E (10 mM Tris-HCl pH 8, 5 mM $MgCl_2$, 100 mM NaCl) at 37°C for 10 min. Complexes were separated on 8% native polyacrylamide gels. This experiment was also done with dsDNA obtained from annealing XV2 with the complementary sequence.

## DNA annealing

Reactions were performed with 200 nM Cy5-labeled Oligo 25 and 200 nM Oligo 26, two 48-nucleotide-long complementary primers described in (*Wu et al., 2008*). Each oligonucleotide was incubated without proteins, with 30 nM RPA or with 134nM Rad51 at 30°C for 5 min, before addition of 40 nM of WT or mutated Rad52. An aliquot of the reaction was collected every 2 min and transferred to stop buffer (20 µM unlabeled Oligo 25, 0.5% SDS, 0.5 mg/ml proteinase K). Samples were separated on 8% native TBE polyacrylamide gels. Fluorescent signals were revealed with a Typhoon 9400 scanner and quantified with the ImageQuant TL software.

## Electron microscopy analysis

For transmission electron microscopy studies, a fraction of the following Rad51 filament formation reactions were used. Standard reactions were done by incubating 15 µM (nucleotides) viral (+) strand of φX174 DNA with 5 µM Rad51 in a buffer containing 10 mM Tris-HCl pH 7.5, 40 mM KCl (or 140 mM KCl when specified), 3 mM $MgCl_2$, 1 mM DTT and 2 mM ATP at 37°C for 3 min. Then, 0.5 µM RPA was added for 5 min. For filament formation in the presence of Rad52, 15 µM φX174 ssDNA was incubated with 1.5 µM saturated RPA at 37°C in the same buffer for 5 min before addition of 5 µM Rad51 and 1 µM Rad52 at 37°C for 20 min. Partial Rad51 filaments range from 1500 to 2500 nm while the size of complete Rad51 filaments is 2800 nm. Srs2 dismantling effect was tested by adding 200 nM Srs2 and 1 mM ATP to the Rad51 filament formation reaction at 37°C for 5 min. Of each reaction, 5 µl was deposited onto a 600 mesh copper grid coated with a thin carbon film, previously activated by glow-discharge in the presence of pentylamine (Merck, France). After 1 min, grids were washed with aqueous 2% (w/v) uranyl acetate (Merck, France) and then dried with ashless filter paper (VWR, France). Observations were carried out using a Zeiss 902 transmission electron microscope in filtered annular dark field mode. Images were acquired with a Veletta digital camera and the iTEM software (Olympus, Soft Imaging Solutions).

### DNA strand exchange reaction

30 μM (nucleotides) viral (+) strand of φX174 DNA were coated first with 3.3 μM RPA by incubation in SEB buffer (42 mM MOPS pH 7.2, 3 mM Mg acetate, 1 mM DTT, 20 mM KCl, 25 μg/ml BSA and 2.5 mM ATP) in a final volume of 12.5 μl at 37°C for 10 min. Rad51 filament formation was initiated by adding 10 μM Rad51 and 15 μM WT or mutated Rad52, or storage buffer as control, at 37°C for 20 min. Then, addition of 30 μM (nucleotides) of PstI-linearized φX174 dsDNA and 4 mM spermidine initiated the strand exchange reaction. After incubation at 37°C for 90 min, samples were deproteinized by addition of 2 μl of 10 mg/ml proteinase K, 5% SDS solution at 37°C for 10 min and analyzed by electrophoresis (0.8% agarose gels in 1x TAE buffer). Standard reactions were done by adding Rad51 prior to RPA. Gels were stained with ethidium bromide and fluorescent signals were imaged with a Typhoon 9400 scanner and quantified with the ImageQuant TL software. The ratio of nicked circular product was calculated as the ratio between the sum of the linear dsDNA substrate and the nicked circular product.

### Challenging Rad51 filament with excess amounts of DNA

Rad52-catalyzed Rad51 filament formation was performed as follow. 165 nM RPA (1:15 nt) was incubated with 2.5 μM of 400nt-long ssDNA (5' end-Cy5-labeled) in SEB buffer (final volume of 10 μl) at 37°C for 10 min. After addition of 1.2 μM Rad51 (1:2 nt) and 165nM Rad52 or Rad52-Y376A (1:15 nt), reactions were incubated at 37°C for 20 min. Then, ΦX174 viral (+) strand (from 5 to 12.5 μM) was added and incubated at 37°C for 30 min. Finally, after fixation with 0.25% glutaraldehyde, 4 μl of 40% sucrose was added to facilitate loading on 0.5% agarose gel. After electrophoresis in 1X TAE buffer at 100V for 1.5 hr, the fluorescent signals were imagined with a Typhoon 9400 scanner.

### Benzonase assay

Rad52-catalyzed Rad51 filaments were formed as described above. Nucleofilaments were then incubated with benzonase (from 0 to 1.25 units) and 2 mM MgOAc at 30°C for 20 min. Reactions were stopped by incubation with 0.5% SDS and 0.5 mg/ml proteinase K at 37°C for 15 min. Deproteinized ssDNA samples were run on 1% agarose gel in 1x TAE buffer at 100 V for 1 hr. Fluorescent signals were revealed with a Typhoon 9400 scanner.

### Measurement of Srs2 ATPase activity

ATPase assays were performed in strand exchange conditions with modifications in protein concentrations. 0.5 μM RPA was first incubated with 5 μM (nucleotides) viral (+) strand of φX174 DNA in 100 μl SEB buffer at 37°C for 10 min. 1.7 μM Rad51 and 0.33 μM Rad52 were then added and samples were incubated at 37°C for 20 min to allow Rad51 filament formation. Standard reactions were done by adding Rad51 prior to RPA. The addition of 50 nM Srs2 and an ATP regeneration system (20 U/ml pyruvate kinase, 20 U/ml lactate deshydrogenase, 0.4 mg/ml NADH) initiated the ATP hydrolysis reaction. Absorbance at 340 nm was measured at 37°C using a microplate spectrophotometer (CLARIO Star, BMG LABTech) every 2 min for 20 min. ATP hydrolysis was expressed as the ratio between the absorbance at 340 nm at a given time and the absorbance measured before Srs2 addition.

## Acknowledgements

We thank the members of the Molecular biology platform (Cigex) of our Institute, Eléa Dizet, Xavier Veaute and Didier Busso for cloning, directed mutagenesis and for RPA, Rad51 and Srs2 purification. We thank Yuen-Ling Chan and Doug Bishop for His-tagged protein purification technical advice. We are grateful to Vincent Géli, Jim Haber, Stefan Jentsch, Stephane Marcand, Richard Kolodner, and Steeve Kowalczykowski, for plasmids and antibodies. We also thank Karine Dubrana, Francis Fabre, Jim Haber, Stephane Marcand, Gérard Mazon and Pablo Radicella for critical and careful reading of the manuscript. We also appreciate the help of Elisabetta Andermarcher with the English editing. EC's laboratory was supported by the Association pour la Recherche sur le Cancer (Projet ARC n° SFI20121205689 and n°PJA 20141201772), La ligue Contre le Cancer (Comité des Hauts-de-Seine, 2015–16) and the Commissariat aux Energies Atomiques et Alternatives (recurrent funding). RG's laboratory was supported by the grant ANR-15-CE11-0008-01 and ELC by the grants ANR-13-BSV8-

0022, Region Ile-de France DIM Nano-K N°IF13012333 and La ligue Contre le Cancer (Comité des Yvelines, 2016–17).

## Additional information

### Funding

| Funder | Grant reference number | Author |
|---|---|---|
| Commissariat à l'Énergie Atomique et aux Énergies Alternatives | Recurrent funding | Raphaël Guerois<br>Eric Coïc |
| Centre National de la Recherche Scientifique | Recurrent funding | Eric Le Cam |
| Fondation ARC pour la Recherche sur le Cancer | SFI20121205689 | Eric Coïc |
| Ligue Contre le Cancer | 2015-16 | Eric Coïc |
| Agence Nationale de la Recherche | ANR-15-CE11-0008-01 | Raphaël Guerois |
| Region Ile-de-France | DIM Nano-K No F13012333 | Eric Le Cam |
| Fondation ARC pour la Recherche sur le Cancer | PJA 20141201772 | Eric Coïc |
| Ligue Contre le Cancer | 2016-2017 | Eric Le Cam |
| Agence Nationale de la Recherche | ANR-13-BSV8-0022 | Eric Le Cam |

The funders had no role in study design, data collection and interpretation, or the decision to submit the work for publication.

### Author contributions

Emilie Ma, Resources, Data curation, Formal analysis, Investigation, Visualization, Methodology; Pauline Dupaigne, Resources, Data curation, Formal analysis, Validation, Investigation, Visualization, Methodology; Laurent Maloisel, Resources, Data curation, Formal analysis, Validation, Investigation, Visualization; Raphaël Guerois, Data curation, Investigation, Visualization, Methodology; Eric Le Cam, Resources, Data curation, Supervision, Funding acquisition; Eric Coïc, Conceptualization, Resources, Data curation, Formal analysis, Supervision, Funding acquisition, Validation, Investigation, Visualization, Methodology, Writing—original draft, Project administration, Writing—review and editing

### Author ORCIDs

Eric Coïc http://orcid.org/0000-0002-9549-8969

### Decision letter and Author response

Decision letter https://doi.org/10.7554/eLife.32744.025
Author response https://doi.org/10.7554/eLife.32744.026

## Additional files

### Supplementary files

• Supplementary file 1. *RAD52* mutations selected by random and directed mutagenesis.
DOI: https://doi.org/10.7554/eLife.32744.022

• Supplementary file 2. *S. cerevisiae* strains.
DOI: https://doi.org/10.7554/eLife.32744.023

### Data availability

All data generated or analysed during this study are included in the manuscript and supporting files.

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
