## [Decision Letter]

Thank you for submitting your article "Rad52 association with Rad51 filaments is crucial to avoid their disruption by the Srs2 DNA translocase" for consideration by *eLife*. Your article has been reviewed by by three peer reviewers, one of whom is a member of our Board of Reviewing Editors, and the evaluation has been overseen by Jessica Tyler as the Senior Editor. The reviewers have opted to remain anonymous.

The reviewers have discussed the reviews with one another and the Reviewing Editor has drafted this decision to help you prepare a revised submission.

Summary:

In yeast, the Rad52 protein has a recombination mediator function to load Rad51 on RPA-coated ssDNA, which facilitates recombination. In contrast, the Srs2 helicase disrupts the Rad51 filaments and served as "anti-recombinase" to remove unscheduled recombination intermediates. In this study, Ma et al., screen for mutants of the Rad52 protein that are able to suppress the MMS sensitivity of Srs2-deficient cells as a way to explore the role of Rad52 in Rad51 filaments in DSB repair and how this relates with Srs2 function. Surprisingly, the authors find mutations in the FAP and YEKF domains of Rad52, which are known to impair Rad52 binding to Rad51, among those suppressors. The study is focused in two of the mutants selected, rad52-P381S and rad52-Y376A with different degrees of suppression. Based on genetic and biochemical analysis of these rad52 mutants, including EM analysis of Rad51 filaments, Ma et al. concluded that Rad51-Rad52 association is important for the protection of Rad51-filament against Srs2 helicase that dismantles Rad51 filament in an ATPase dependent manner.

The Rad52's role in a post-assembly step of Rad51 filament by the filament stabilization is an important conclusion, since the role of Rad52 to promote the formation of Rad51 filament as a mediator in pre-assembly step is believed to be a main function of Rad52 for Rad51-dynamics in the recombination. Indeed, a similar role to protect Rad51 filament against Srs2 helicase has been reported to Rad55-57 complex by W. Heyer's group. The authors also provide evidence favoring that Rad52-Rad51 interaction promotes the mediator function, and that the role of the 51-52 interaction goes beyond protection against Srs2, which should be properly discussed. Although most results are convincing and the conclusions relevant, few issues should be addressed to get further mechanistic insight on how Rad52 stabilizes Rad51-filaments on ssDNA.

Essential revisions:

- Different results in this paper argue against the conclusion that "the interaction between Rad52 and Rad51 is not required for Rad51 filament assembly, but for preventing Rad51 filament disassembly by Srs2": (a) the SSA assay (Figure 4A) showing that cells lacking srs2 are impaired in SSA because Rad51 filaments are more stable, and that P381S and Y376A Rad52 mutants (in srs2Δ) rescues SSA because they reduce the stability/formation of Rad51 filaments, demonstrates that the mutations have a phenotype in the absence of Srs2, and the effect of the Rad51-Rad52 interaction is therefore not specific to protection from Srs2; (b) the Rad52-Y376A mutant accumulates more RPA and less Rad51 as shown by ChIP (Figure 5), and is severely impaired in strand exchange (Figure 6C), consistent with a Rad51-Rad52 interaction for the mediator role, and (c) if the interaction of Rad51-Rad52 were not important for HR but to prevent filament disassembly by Srs2, it would be necessary to explain why Rad52 P381S and Y376A mutants have such a strong phenotype without Srs2 (Figure 4B), The manuscript should be therefore rewritten to acknowledge these points.

- Do the Rad51 Y376A or P381S mutations affect interaction with the Rad51 filament? This could be tested by performing pulldowns with biotinylated oligonucleotides pre-bound with Rad51. The assay should also be used to directly test the effect of Rad52 on the protection of Rad51 filaments from disruption by Srs2 (e.g., see Figure 2 in Liu et al., 2011) by preparing Rad51 filaments on ssDNA, without or with Rad52, and challenging it with Srs2 or e.g. DNAseI. This will dissect whether the effect is Srs2-specific, or whether Rad52 stabilizes the Rad51 nucleoprotein filament.

- Most interactions of Rad52 mutant proteins with Rad51 had been tested in the presence of 140mM NaCl (Figure 2). However, EM analysis (in the presence of 40 mM KCl) clearly showed that Rad52 mutant proteins associate with Rad51 filaments to the similar extent to the wild-type Rad52 protein (Figure 6B). This discrepancy should be clarified by doing IP or pull-down experiments under the same condition for EM analysis and vice versa.

- Although Rad52 stabilizes Rad51 filaments against the translocase activity of Srs2, it is important to show biochemically Rad52 indeed stabilizes Rad51 filament even in the absence of Srs2 by examining the effect of Rad52 on Rad51 filament by high salt attack or challenging by excess amounts of ssDNAs (with ATP turnover). This can be easily tested by analyzing Rad51 filaments by EM or by measuring ATPase activity of Rad51. This approach would also reveal a mechanism on how Rad52 stabilizes Rad51 even in the absence of Srs2.

- Although Rad52 does not inhibit ATPase activity of Srs2, it is important to show that Rad52 by itself does not inhibit the helicase activity of Srs2, given that Rad52 and Srs2 interacts with each other.

- The ChIP-qPCR shown in Figure 5 for Rad52 seems to have been performed with a Rad52-FLAG. Since the authors are analyzing two specific mutants for Rad52, it seems that they are detecting is a WT copy of Rad52 carrying FLAG, so that the mutant rad52 proteins are not detected. Those experiments need to be explained better how they were performed. If the authors are detecting only wt Rad52, this needs to be discussed. It is unclear what would be meaning of those data because authors are not detecting the mutant Rad52, subject of study. The results as such are confusing and needs to be clarified.

- It is a bit surprising to use a SSA system that does not allow for identification of recombinants by a genetic selection of Leu2+ recombinants. However, authors should try to determine recombination via qPCR in the SSA assay used in Figure 4. It would be nice to see whether in the srs2 rad52-P381S and srs2 rad52-Y376S mutants the frequency of recombination accompanies the data of survival.

---

## [Author Response]

Summary:In yeast, the Rad52 protein has a recombination mediator function to load Rad51 on RPA-coated ssDNA, which facilitates recombination. In contrast, the Srs2 helicase disrupts the Rad51 filaments and served as "antirecombinase" to remove unscheduled recombination intermediates. In this study, Ma et al., screen for mutants of the Rad52 protein that are able to suppress the MMS sensitivity of Srs2-deficient cells as a way to explore the role of Rad52 in Rad51 filaments in DSB repair and how this relates with Srs2 function. Surprisingly, the authors find mutations in the FAP and YEKF domains of Rad52, which are known to impair Rad52 binding to Rad51, among those suppressors. The study is focused in two of the mutants selected, rad52-P381S and rad52-Y376A with different degrees of suppression. Based on genetic and biochemical analysis of these rad52 mutants, including EM analysis of Rad51 filaments, Ma et al. concluded that Rad51-Rad52 association is important for the protection of Rad51-filament against Srs2 helicase that dismantles Rad51 filament in an ATPase dependent manner.The Rad52's role in a post-assembly step of Rad51 filament by the filament stabilization is an important conclusion, since the role of Rad52 to promote the formation of Rad51 filament as a mediator in pre-assembly step is believed to be a main function of Rad52 for Rad51-dynamics in the recombination. Indeed, a similar role to protect Rad51 filament against Srs2 helicase has been reported to Rad55-57 complex by W. Heyer's group. The authors also provide evidence favoring that Rad52-Rad51 interaction promotes the mediator function, and that the role of the 51-52 interaction goes beyond protection against Srs2, which should be properly discussed. Although most results are convincing and the conclusions relevant, few issues should be addressed to get further mechanistic insight on how Rad52 stabilizes Rad51-filaments on ssDNA.

Thank you for the nice summary. We present below our arguments to defend the three major points of the paper:

1) Rad52-Rad51 interaction is responsible for “Rad51 filament toxicity” in Srs2deficient cells because disruption of Rad51 binding domain in Rad52 restores MMS resistance in Srs2-deficient cells.

2) Rad52-Rad51 interaction is not essential for Rad51 filament assembly.

3) In Srs2-proficient cells, the Rad52/Rad51 interaction protects the Rad51 filament against disassembly by Srs2.

We understand that our arguments were not explicit enough and consequently, we have made substantial modifications to the manuscript that are shown in blue in the revised text. We decided to add some technical descriptions of our experiments in an annex to this letter, to ease the reading of our answers to your comments.

Essential revisions:- Different results in this paper argue against the conclusion that "the interaction between Rad52 and Rad51 is not required for Rad51 filament assembly, but for preventing Rad51 filament disassembly by Srs2": (a) the SSA assay (Figure 4A) showing that cells lacking srs2 are impaired in SSA because Rad51 filaments are more stable, and that *P381S* and *Y376A* Rad52 mutants (in srs2Δ) rescues SSA because they reduce the stability/formation of Rad51 filaments, demonstrates that the mutations have a phenotype in the absence of Srs2, and the effect of the Rad51-Rad52 interaction is therefore not specific to protection from Srs2;

We agree with the statement in (a) that there is a role for Rad52 that is not specific to the protection from Srs2. As stated in the last paragraph of the Introduction, we found previously that Rad52 is at the origin of Srs2-deficient cell MMS sensitivity, possibly through the interaction with Rad51 nucleoprotein filaments. We hypothesized that this association confers an excessive stability to Rad51 filaments in Srs2-deficient cells. We have defined such filaments as toxic, because they are responsible for MMS sensitivity in Srs2-deficient cells. In the present work, we designed a genetic screen to identify Rad52 mutations that suppress MMS sensitivity and consequently Rad51 filament toxicity in Srs2-deficient cells. In the first part of the Results section, we report that with this screen, we identified mutations in the Rad51 binding domain. This finding confirmed that the interaction between Rad52 and Rad51 is responsible for Rad51 filament toxicity in Srs2-deficient cells. Therefore, we conducted a mutational analysis of the Rad51 binding domain (second and third part of the Results section) to confirm this link. The finding that the stronger suppressor phenotype can be attributed mostly to mutations that affect the interaction between Rad52 and Rad51 confirmed this link. In conclusion, Rad52 is responsible for the death of Srs2-deficient cells following exposure to MMS or γ-rays, because of its interaction with Rad51.

As underlined by the reviewers, we found that the Rad52-P381S and Rad52-Y376Amutants also rescue SSA in Srs2-deficient cells. In the first part of the Discussion section, we propose that the interaction between Rad52 and Rad51 is responsible for the toxicity of Rad51 filaments at the HO-induced DSB site in Srs2-deficient cells. Therefore, Rad52 would have at least two functions that are Srs2-independent: (i) displacement of RPA to mediate Rad51 filament formation, and (ii) stabilization of Rad51 filaments, which is the cause of their toxicity in the absence of Srs2. Nevertheless, we agree that we have not underlined enough the involvement of Rad52 in Rad51 filament toxicity in several parts of the paper, which could be misleading. For clarification, we modified the Abstract, the last paragraph of the Introduction section, and the Results section. We cannot introduce this notion in the Title because of character restrictions, but we added the idea that Rad52-Rad51 is facultative to filament formation, which is another conclusion about the Srs2-independent functions of Rad52 (see next section).

(b) The Rad52-Y376A mutant accumulates more RPA and less Rad51 as shown by ChIP (Figure 5), and is severely impaired in strand exchange (Figure 6C), consistent with a Rad51-Rad52 interaction for the mediator role.

About point (b), the reason we claim that the Rad52-Rad51 interaction is not essential to the formation of Rad51 filaments is based on several results we think it is important to summarize here before considering the ChIP experiment addressed by the reviewers. As explained in the Abstract and in the first two parts of the Results section, the double mutants *rad52-P381S srs2*∆ and *rad52-Y376A srs2*∆ are not sensitive to MMS or γ-ray irradiation (Figure 1). This shows thatMMS and γ-ray sensitivity in the *rad52-P381S* and *rad52-Y376A* single mutants are dependent on Srs2 activity. When Srs2 is depleted, the loss of interaction between Rad52 and Rad51 (as observed by co-IP and pull-down experiments) does not impact HR sufficiently to impair repair. Actually, measurement of γ-ray-induced HR frequencies in the recombination system between heteroalleles of the *ARG4* gene (Figure 3 and subsection “Rad52 interaction with Rad51 is dispensable for gene conversion”) shows that HR is not affected in the mutants. In addition, we observed that repair by gene conversion of a HO-induced DSB is not significantly affected by *rad52-P381S* and *rad52-Y376A* (now in Figure 3). Altogether, these results strongly suggest that both Rad52 mutants, despite the weak interaction with Rad51, form Rad51 filaments. This notion is now added in the Title. It is true that other regions that would bind to Rad51 (even very weakly, according to the faint interaction we observed with the *rad52-Y376A* mutant even when pull-down assays were performed with 40mM NaCl) could help the loading of the protein. We added a comment in subsection “The *rad52-Y376A* and *rad52-P381S* mutations impact the interaction between Rad52 157 and Rad51” to underline this point. The domains that could be involved in the association of Rad52-Y376A with Rad51 filaments are discussed in the Discussion section. Nevertheless, the Srs2-independent role of Rad52 is related to the stability and not to the formation of Rad51 filaments.

Concerning the ChIP experiments (Figure 5), it is true that the *rad52-Y376A* mutant accumulates more RPA and less Rad51, as it was explained in the Results section. Two hypotheses can be made from this observation. The first one is that *rad52-Y376A* is not able to normally assemble Rad51 filaments. The second proposes that according to the genetic relationship we observed between *srs2*∆ and *rad52-P381S* and *rad52-Y376A*, Srs2 removes more efficiently Rad51 filaments because they are less protected by the Rad52 mutants. We observed that in *rad52P381S srs2*∆ and *rad52-Y376A srs2*∆ double mutants, Rad51 is recruited at similar levels as in *srs2*∆ cells. This demonstrates, first, that Srs2 activity is responsible for the reduction in Rad51 recruitment in Rad52 mutants and, secondly, that both Rad52 mutants can form Rad51 filaments as efficiently as in WT, if Srs2 is depleted. To make this conclusion clearer, we modified the text in subsection “Rad52 interaction with Rad51 is dispensable for Rad51 filament formation at a HO-212 induced DSB but is essential for preventing Rad51 filament disassembly by Srs2” to develop this demonstration better.

As mentioned by the reviewers, it is true that the Rad52-Y376A mutant is severely impaired in strand exchange (now Figure 5C). This point is mentioned in the Results section. However, the reviewers have not compared this result with our observation that Rad52-Y376A forms Rad51 filaments on RPA-covered ssDNA (now Figure 5A). The mediator activity of the mutant protein is robust because 15 min incubation of Rad52-Y376A with Rad51 on RPA-covered ssDNA leads to the formation of 70% of complete Rad51 filaments, compared to 100% with the WT protein and 25% in the absence of Rad52 in the reaction. We cannot explain why this large amount of complete Rad51 filaments cannot achieve strand invasion and we discussed this point in the Discussion section. However, it is important to keep in mind that, as discussed above, Rad52-Y376A forms enough Rad51 filaments in vivo to ensure proper DNA repair after MMS and γ-rays irradiation or HO-induced DSB formation. We also added to the Discussion section that the Rad51 paralogues Rad55/Rad57 and SHU can compensate the effect of the mutation in vivo.

(c) If the interaction of Rad51-Rad52 were not important for HR but to prevent filament disassembly by Srs2, it would be necessary to explain why Rad52 *P381S* and *Y376A* mutants have such a strong phenotype without Srs2 (Figure 4B), The manuscript should be therefore rewritten to acknowledge these points.

About point c), the strong phenotype of the double mutants *rad52-P381S srs2*∆ and *rad52-Y376A srs2*∆ observed in Figure 4B cannot be attributed to an inability of the Rad52 mutant proteins to assemble functional Rad51 filaments. As explained in the Results section, the new Figure 3B shows that the *rad52-P381S* and *rad52-Y376A* mutant cells are not defective for the repair of a HO induced DSB at the *MAT* locus by gene conversion. Gene conversion absolutely requires the formation of Rad51 filaments to invade the donor sequence. Therefore, the only possible explanation for the finding that these two Rad52 mutants show only a slight decrease in viability after HO cut (not statistically significant for the *rad52-P381S* and barely significant for the *rad52-Y376A* mutant) is that Rad51 filaments are formed at the DSB site, which is observed at another locus in our ChIP experiments. However, viability of Srs2-deficient cells is very low after HO-induction in the gene conversion system. As explained in the Results section, this strong lethality has been described before (Ira et al., 2003). In Srs2-deficient cells, a large number of recombination events happening through the SDSA pathway are not resolved properly, leading to cell death. It is this defect that also leads to lethality in cells harboring the double *rad52-P381S srs2*∆ and *rad52-Y376A srs2*∆ mutations. If *rad52-Y376A srs2*∆ double mutant cells were sensitive to the HO-induced DSB because Rad52-Y376A cannot assemble Rad51 filaments, then, *rad52-Y376A* cells also would be sensitive to DSB formation. This is not the case. Moreover, the *rad52-P381S* or *rad52-Y376A* mutants do not reduce γ-ray-induced HR, as measured in the *ARG4* system (Figure 3).

In summary, it seems that the way we presented our data was confusing because major conclusions were not understood. Consequently, we introduced major changes in the organization of the Results section. First, at the end of the second chapter, we explained the assumptions that can be made from our first results. Second, we grouped our two gene conversion assays together (*ARG4* and *MAT*) because they both show that the *rad52-P381S* and *rad52-Y376A* mutations do not affect gene conversion (Figure 3). Then, we decided to present directly the ChIP experiments because they confirm at the molecular level that the *rad52-P381S* and *rad52-Y376A* mutant cells can assemble Rad51 filaments in Srs2deficient cells. Finally, we present the results obtained with the SSA system. By presenting the ChIP experiments before the SSA system results, it is easier for the reader to understand that the rescue of the *srs2*∆ phenotypes by the Rad52 mutations is not related to a default in Rad51 filament formation, but more certainly to the destabilization of Rad51 filaments in vivo by the loss of Rad52-Rad51 interaction.

- Do the Rad51 *Y376A* or *P381S* mutations affect interaction with the Rad51 filament? This could be tested by performing pulldowns with biotinylated oligonucleotides pre-bound with Rad51. The assay should also be used to directly test the effect of Rad52 on the protection of Rad51 filaments from disruption by Srs2 (e.g., see Figure 2 in Liu et al., 2011) by preparing Rad51 filaments on ssDNA, without or with Rad52, and challenging it with Srs2 or e.g. DNAseI. This will dissect whether the effect is Srs2-specific, or whether Rad52 stabilizes the Rad51 nucleoprotein filament.

In the original version of the manuscript, we counted the number of WT and mutated Rad52 molecules observed by EM on in vitro assembled Rad51 filaments (previous Figure 6B). This experiment shows a significant decrease in the average number of mutated Rad52 associated with the filaments (now Figure 5B). We now present this result in a clearer way in the text (subsection “in vitroanalysis of Rad52 activities when interaction with Rad51 is altered”) and we changed the representation of our data on the graph.

We agree that pulldown assays could have provided useful information on Rad52 effect on Rad51 filaments. However, a series of technical problems did not allow us to obtain results, despite several attempts to optimize the assay (see annex for technical details). Consequently, we assessed the role of Rad52 on Rad51 filaments formed on 400nucleotides (nt)-long ssDNA by using a gel shift assay we described before (Esta et al., 2013), to avoid the use of streptavidin beads. In fact, we did several preliminary experiments before submitting the original version of the manuscript, but we were not convinced that this assay would allow testing correctly the role of Rad52 when associated with Rad51 filaments. Indeed, we observed previously by EM that only 10% of complete Rad51 filaments were associated with more than one Rad52 spot (Esta et al., 2013). Therefore, we worried that experiments designed to study Rad52 effect on Rad51 filaments with this assay would be difficult to interpret. Anyway, following your suggestions, we decided to carry out several experiments to test whether we could obtain some positive data. These results are described below in the section concerning Rad51 filament stability because several questions overlap.

- Most interactions of Rad52 mutant proteins with Rad51 had been tested in the presence of 140mM NaCl (Figure 2). However, EM analysis (in the presence of 40 mM KCl) clearly showed that Rad52 mutant proteins associate with Rad51 filaments to the similar extent to the wild-type Rad52 protein (Figure 6B). This discrepancy should be clarified by doing IP or pull-down experiments under the same condition for EM analysis and vice versa.

This is a very good point. Consequently, we performed EM analysis in the presence of 140mM NaCl and pull-down assays in the presence of 40mM NaCl. We found that a light interaction could be observed by pull-down assay at this low salt concentration, and that Rad52-Y376A still interacts with Rad51 filaments at high salt concentration (43% of WT). These results were added to the manuscript and discussed.

- Although Rad52 stabilizes Rad51 filaments against the translocase activity of Srs2, it is important to show biochemically Rad52 indeed stabilizes Rad51 filament even in the absence of Srs2 by examining the effect of Rad52 on Rad51 filament by high salt attack or challenging by excess amounts of ssDNAs (with ATP turnover). This can be easily tested by analyzing Rad51 filaments by EM or by measuring ATPase activity of Rad51. This approach would also reveal a mechanism on how Rad52 stabilizes Rad51 even in the absence of Srs2.

To answer these questions, as well as the previous remarks concerning the potential effect of the *rad52-P381S* and *rad52-Y376A* mutations on Rad52 interaction with Rad51 filaments, we analyzed Rad51 filament formation on 400nt-long ssDNA using a gel shift assay. First, we showed that Rad52-Y376A could assemble Rad51 filaments despites its poor association with Rad51, confirming our EM observation on PhiX174 DNA. This is now shown in Figure 5—figure supplement 2A and is also detailed in subsection”in vitroanalysis of Rad52 activities when interaction with Rad51 is altered”.

We agree that it is important to test whether Rad52 can stabilize Rad51 filaments in vitro. However, this cannot be addressed simply by high salt attack because we published earlier that Rad52 is displaced at the low salt concentrations to which the Rad51 filament is not sensitive (Esta et al., 2013). This was confirmed here by the EM analysis of Rad51 filament formation. WT Rad52 is already strongly displaced from filaments in the presence of 150mM NaCl. Therefore, we decided to challenge Rad51 filaments by using excess amounts of ssDNA. Unfortunately, this experiment cannot be done by EM, because the competing DNA hinders the spreading of Rad51 filaments on carbon grids. Consequently, we set up experiments using our previously described gel shift assay (Esta et al., 2013). We challenged filaments with an excess of ssDNA (we used PhiX174 ssDNA at concentration 2 to 5fold higher than that of 400nt-long ssDNA). We found no difference in the stability of Rad51 filaments assembled with WT Rad52 or Rad52-Y376A (Figure 5—figure supplement 2A). We also performed experiments with benzonase (Figure 5—figure supplement 2B), but again, we did not see any difference between Rad51 filaments assembled with Rad52 or Rad52-Y376A. As our EM observations show that only 10% of Rad51 filaments are associated with more than one Rad52 spot, it is difficult to reach a conclusion. We now explain that our result might indicate that Rad52 cannot stabilize Rad51 filaments or that there is not enough Rad52 associated with Rad51 in these experiments. It is important to remember that Rad52 concentration cannot be increased in these experiments because it results in an inhibition of Rad51 filament formation. We also tested Rad51 filament stability by measuring Rad51 ATPase activity using the assay described in Li et al., (2007). No ATPase activity difference was observed between Rad51 filament formed with WT or mutated Rad52. Therefore, we found no way to address the question of the stabilization of Rad51 filaments by Rad52 by biochemical assays. We present this problem in the text, and this justifies the use of PhiX DNA, which is larger (5 kb), to test the effect of the Rad52 mutations on Rad51 filament formation and protection against Srs2. Effectively, the EM observation of Rad51 filaments formed with the mutated proteins shows undoubtedly that Rad52-Y376A can assemble Rad51 filaments and that Rad52 protects Rad51 filaments from Srs2 translocase activity.

Concerning the effect of the Y376A and P381S mutations on Rad52 association with the Rad51 filament, we cannot address this question with the gel shift assay. Previously, we showed by western blot analysis of the different nucleoprotein complexes separated on agarose gels that Rad52 could be detected on complete Rad51 filaments (Esta et al., 2013). We could do this using an anti-Rad52 polyclonal antibody provided by Stephen Jentsch. Unfortunately, this antibody is no longer available. We tested recently a commercial anti-Rad52 antibody (Santa Cruz), but we found that it cross-reacts with Rad51. We also used an anti-FLAG antibody because Rad52 is FLAG-tagged in our experiments. This antibody revealed Rad52-FLAG in cell extracts analyzed by western blotting and was used in our co-IP and ChIP experiments. However, this antibody cannot detect Rad52 associated with Rad51 filaments in gel shift assays. The tag might be hidden in the context of the Rad52/Rad51/ssDNA complex, or Rad52 C-terminus could suffer proteolysis in vitro. Altogether, these results are disappointing, but we think that our EM study of Rad51 filament formation on PhiX174 DNA shows the effect of *rad52-Y376A* and *rad52P381S* on the association of Rad52 with Rad51 filaments (now Figure 5B). We also used this gel shift assay to try to confirm the Rad52 inhibitory effect on Srs2 activity observed by EM (now Figure 5D). However, we failed. Again, this might be caused by the low number of Rad52 molecules associated with filaments assembled on 400nt-long ssDNA. Therefore, there might be a minimum size of Rad51 filament required to observe the effect of Rad52 on Srs2. We cannot exclude either that the PhiX174 DNA circular nature is important for observing this inhibition in vitro. These hypotheses will require to be tested in a complete study of the inhibition of Srs2 by Rad52. We decided not to include these results in the manuscript at this time.

- Although Rad52 does not inhibit ATPase activity of Srs2, it is important to show that Rad52 by itself does not inhibit the helicase activity of Srs2, given that Rad52 to Srs2 interacts with each other.

We faced a technical problem also with this experiment (see annex) and decided not to include it in our manuscript.

- The ChIP-qPCR shown in Figure 5 for Rad52 seems to have been performed with a Rad52-FLAG. Since the authors are analyzing two specific mutants for Rad52, it seems that they are detecting is a WT copy of Rad52 carrying FLAG, so that the mutant rad52 proteins are not detected. Those experiments need to be explained better how they were performed. If the authors are detecting only wt Rad52, this needs to be discussed. It is unclear what would be meaning of those data because authors are not detecting the mutant Rad52, subject of study. The results as such are confusing and needs to be clarified.

These experiments were done in haploid cells. Therefore, there is only one of Rad52-FLAG that is expressed (WT or mutant). The protein detected is WT or mutant and there is never co-expression of WT and mutant protein. This has been clarified in the text (subsection “Rad52 interaction with Rad51 is dispensable for Rad51 filament formation at a HO-induced DSB but is essential for preventing Rad51 filament disassembly by Srs2”).

- It is a bit surprising to use a SSA system that does not allow for identification of recombinants by a genetic selection of Leu2+ recombinants. However, authors should try to determine recombination via qPCR in the SSA assay used in Figure 4. It would be nice to see whether in the srs2 rad52-P381S and srs2 rad52-Y376S mutants the frequency of recombination accompanies the data of survival.

A system that allows the formation of LEU^+^ recombinants would allow the formation of prototrophes by gene conversion. The SSA system we used was designed by (Vaze et al., 2002) to allow the repair of a HO-induced DSB only by SSA and not by a classical gene conversion mechanism (DSBR). This is why repair events cannot be selected. As described in the 2002 paper, the amount of product in Srs2-deficient cells does not reflect cell survival. The amount of product is surprisingly high despite the low viability of Srs2-deficient cells. We repeated this experiment by using PCR with primers that specifically amplify the repair products and showed that the amount of product was roughly the same in WT and *srs2*∆ cells. We also found the same for *rad52-Y376A* and *rad52-Y376A srs2*∆ cells. Because this was described earlier for *srs2*∆, and was not changed by *rad52-Y376A*, we chose not to add it in the paper.

**Annex**

Pull-down assay

When trying to set up the pull-down assay, we faced several problems that are, in part, related to the complexity of Rad51 filament formation in vitro. As described by Esta et al. (2013), the addition of Rad51 and Rad52 to a 400nt-long ssDNA resulted mostly in partial Rad51 filament formation, and only 4% of the ssDNA molecules were completely covered by Rad51, as indicated by the EM results. RPA is essential to form complete Rad51 filaments. Additionally, the stoichiometry of RPA, Rad51, Rad52 and ssDNA has to be very precise to allow complete Rad51 filament formation. Increasing Rad52 concentration results in an inhibition of Rad51 filament formation, which makes difficult to test the effect of Rad52 on Rad51 filament stability or on Srs2 activity. We also observed previously by EM (Esta et al., 2013) that only 10% of complete Rad51 filaments formed on a 400nt-long ssDNA is associated with more than one Rad52 spot. This might decrease the chance to observe an effect of Rad52 on Rad51 filaments assembled on small molecules.

Therefore, first, we tried to assemble complete Rad51 filaments on a 60mer biotinylated oligonucleotide. The major problem we encountered was the high unspecific association of Rad51 with the streptavidin-coated magnetic beads (Invitrogen). By trying different conditions, we could at best reduce Rad51 unspecific binding to 20%; however, in these conditions of salt concentration and temperature, Rad52 mediator activity could not be observed (i.e., there was as much Rad51 retained on beads coated with RPA-covered ssDNA in the presence or in the absence of Rad52). Additionally, because the affinity of Rad52 for naked DNA is high, it is impossible to distinguish DNA-bound Rad52 from Rad51 filament-bound Rad52 with this technique.

Then, to reduce the signal/background ratio, we used a 400nt-long ssDNA that shares homology with the biotinylated oligonucleotide to bind it to the streptavidin beads. Unfortunately, we observed high unspecific binding of the 400nt-long ssDNA to the beads in the absence of the oligonucleotide. We varied the salt concentration in the buffer used to coat the beads with the 60mer oligonucleotide. Yet, at best we still found 30% of unspecific binding of the 400nt-long ssDNA to the beads. We considered that we could not proceed on Rad51 filament formation with such a high background.

Srs2 helicase activity assay

We used a 200bp dsDNA substrate with a 400nt-long 3’ overhang, resulting from the pairing of a 600nt ssDNA with a complementary 200nt ssDNA. Srs2 must be loaded on ssDNA to be able to unwind dsDNA. It cannot unwind a blunted end dsDNA. Srs2 displays a poor helicase activity on this substrate. As described previously by several laboratories (REF?), Srs2 helicase activity is strongly enhanced by Rad51 pre-coated on ssDNA. To test the effect of Rad52 on Srs2 helicase activity, we formed Rad51 filaments by adding Rad52 and Rad51 on the substrate pre-coated with saturating amount of RPA. We found that Rad52 does not reduce Srs2 helicase activity. However, our controls show that saturating amount of RPA on the 3’ tail stimulates Srs2 helicase activity as efficiently as Rad51, thus preventing us to reach a valid conclusion.